# A single mode of population covariation associates brain networks structure and behavior and predicts individual subjects' age

Brent C. McPherson[1] & Franco Pestilli [1,2✉]

Multiple human behaviors improve early in life, peaking in young adulthood, and declining thereafter. Several properties of brain structure and function progress similarly across the lifespan. Cognitive and neuroscience research has approached aging primarily using associations between a few behaviors, brain functions, and structures. Because of this, the multivariate, global factors relating brain and behavior across the lifespan are not well understood. We investigated the global patterns of associations between 334 behavioral and clinical measures and 376 brain structural connections in 594 individuals across the lifespan. A single-axis associated changes in multiple behavioral domains and brain structural connections ($r = 0.5808$). Individual variability within the single association axis well predicted the age of the subject ($r = 0.6275$). Representational similarity analysis evidenced global patterns of interactions across multiple brain network systems and behavioral domains. Results show that global processes of human aging can be well captured by a multivariate data fusion approach.

[1] Department of Psychological and Brain Sciences, Indiana University Bloomington, Bloomington, IN, USA. [2] Present address: Department of Psychology, The University of Texas at Austin, Austin, TX, USA. ✉email: pestilli@utexas.edu

Understanding human aging and its progression in health and disease have become a critical need due to the rapidly aging world population[1,2]. Brain aging and the associated cognitive decline negatively impact society by reducing the independence of individuals in the population, with substantial costs associated with increased needs for long-term support or treatment[3–7]. Costs associated with an aging population are only expected to increase over the next few years, given the increase in life expectancy[8–12]. As a result of this looming demographic shift, improving prodromal identification of individuals at risk versus individuals subject to normal aging in large human populations is becoming a priority[13–17].

The normative human aging process is accompanied by behavioral changes in performance in both cognitive and perceptual tasks across the lifespan[18–20]. Early work focussed on measuring cognitive and perceptual aging with deep characterization of a few behavioral tasks[21–24]. A staggering amount of evidence has been reported on the effect of aging to human cognition and perception. For example, language processing and letter perception deteriorate with age[25–27]. At the same time, visual contrast sensitivity, motion detection, recognition, and iconic memory also decay across the lifespan[28–32]. Yet, this is not always the case as resilience to the effect of aging has been reported in a few behavioral domains such as language[26,27,33] (e.g., vocabulary size) and emotion[34,35].

Correspondingly with cognitive aging, brain aging is associated with both changes to neuronal structures and function[21,36–42]. Primary examples of changes to brain structures consist of hippocampal volume reduction[43,44], cortical thinning[45,46], and ventricular expansion[41]. At the same time, multiple examples of changes in brain functional activity have been reported[47,48]. For example, the brain hemodynamic response changes across age[49–51], prefrontal cortex activity changes during a variety of tasks[47,52,53], such as attentional control[54,55], inhibition[54,56], and executive control[19]. Finally, changes to the white-matter tissue properties across the lifespan have been reported[57–64]. These studies painted a comprehensive picture of how aging affects human behavior due to alterations to either brain function and structure across multiple tissue types.

As of today, much attention has been devoted to the characterization of the changes in individual cognitive tasks and brain systems as a result of aging[21,51,65]. This approach has helped tremendously in developing an understanding of the mechanisms involved in individual cognitive functions[66–69]. Yet, a few shortcomings have been noted with single-task approaches[70,71]. For example, limits to the definition of psychological constructs[36,72], and the co-involvement of networks of brain systems in supporting individual psychological domains[62,73–75] hinder our ability to assign a one-to-one correspondence between brain systems and psychological constructs or tasks. For example, challenges in isolating cognitive processes[76–79], resulted in critiques to the very definition of psychological constructs[72]. Furthermore, studying a few constructs at a time may be also limited because multiple processes coexist within individual brain areas, as early as in the sensory systems[80,81].

Besides the mappings between a few functions and brain systems, the global patterns of change in the brain network connectivity associated with human aging, behavior, and cognition remain uncharted. To advance understanding of normative aging, we integrated multivariate[38,40,76,82–84] and network neuroscience[38,85–88] methods to develop a many-to-many map between behavioral domains and brain network systems. A reproducible data preprocessing pipeline was developed for the current work and made freely available as a web service on brainlife.io[89]. The pipeline was used to process data from a large sample of healthy adults (594 subjects, 18–88 years; ref. [90]). Canonical Correlation Analysis (CCA; ref. [83]) was then used to associate multiple behavioral measures (tasks and scales) with the connectivity properties of structural brain networks across age[73,91]. A single CCA axis of covariation successfully mapped brain networks to behavioral features. Critically, the CCA axis was associated with the individual participants' age indicating that a coherent pattern of degradation affects both brain networks and behavior. Finally, representational similarity analysis[92] applied to the cross-validated CCA factors determined which global factors in multiple brain networks[93] and behavioral domains[90] jointly associated with predicting age.

## Results

The goal was to estimate if the structural connectivity properties of human brain networks are associated with human behavior measured from tasks, questionnaires, and scales collected across the lifespan (18–88 years, Supplementary Fig. 1). To do so, data from the Cambridge center for Ageing and Neuroscience was used, the dataset is hereafter referred to as CAN[90]. The CAN dataset contains a deeply phenotyped cohort of cognitively healthy individuals (594 used here for all analyses; see the "Methods" section for inclusion criteria) evenly sampled across the lifespan (100 subjects per decade). We used 388 behavioral scores from 33 assessments published with the CAN dataset. These scores consisted of either reaction times, accuracy, or performance scales–see example histograms of each type of score plotted across age are shown in Fig. 1a and Supplementary Fig. 1. A total of 334 normalized behavioral measures were extracted and utilized for all subsequent analyses (140 reaction time, 188 accuracy, and 6 assessments; see "Methods": Performing the CCA). The CAN behavioral measures were originally mapped into five behavioral domains[90]: attention, language, memory, motor, and emotion. In addition to these, scores from social and clinical assessments were organized into two additional domains for a total of 7 behavioral domains (Fig. 1a).

Whole-brain structural networks were estimated using diffusion-weighted magnetic resonance imaging (dMRI) data from 594 subjects[90]. Neuroimaging data were processed using an automated and reproducible pipeline using brainlife.io (see Table 1; ref. [89]). Figure 1 panels b–d show a representative network, as well as the Human Connectome Project Multi-Modal Parcellation cortical (HCP-MMP v1.0; ref. [94]) and subcortical[95] atlases used for network neuroscience data generation. A total of 376 regions (366 cortical and 10 subcortical) were used to generate each network. The dMRI data was processed for artifact removal and fiber model fitting using a recent robust method (see ref. [96] and bl.app.68). Whole-brain tractograms were generated using a novel pipeline called Reproducible Anatomically Constrained Ensemble Tracking (RACE-Track; see bl.app.101), which integrates two established methods[97,98]. RACE-Track tractograms were combined with the regions of interest from the two atlases (see bl.app.23) to build individual subjects' connectivity matrices. Connections present in less than half of the subjects were eliminated (see also Supplementary Fig. 2a; ref. [99]). Connection density ($C_d$, see Eq. (1), Fig. 1b, generated by web service bl.app.121; ref. [100]) was used as network edge weight. The 70,500 $C_d$ estimates were reduced to 376 node-degree estimates used for all subsequent analyses (see Supplementary Fig. 2a[91]).

**Changes in network neuroscience and behavioral measures across the lifespan.** The relationship between networks' node degree and performance in behavioral tasks and assessments was explored across the lifespan (Fig. 1e, f and Supplementary Fig. 1). The goal was to ensure replication of known trends established in the literature. Changes in both network and behavioral measures were observed—especially starting at the fourth-decade Fig. 1e, f[57].

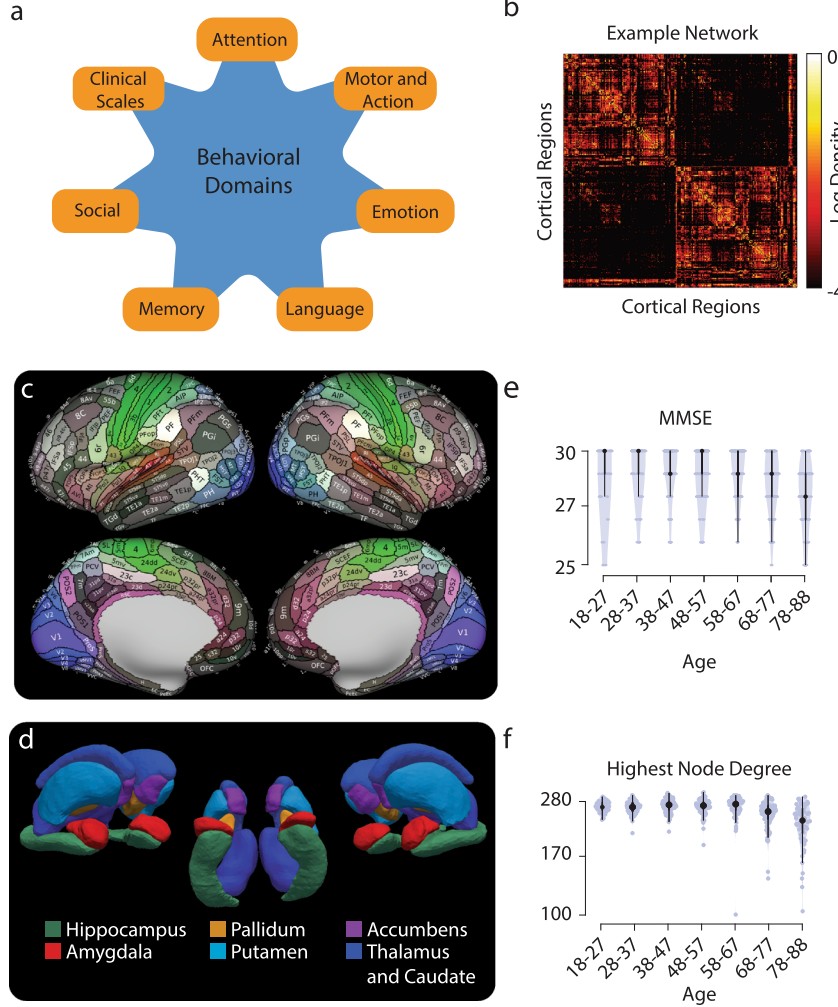

**Fig. 1 Main findings. a** The behavioral domains. A simple graphic displaying the different behavioral domain labels is displayed. **b** An example network. This is an example of an individual's network that is created for analysis. Each row and column represents a cortical region while the color represents the log scale of the density of the connection between nodes (weaker connections are black, stronger connections are yellow). **c** The cortical regions used to create the network structures. This figure, taken from Glasser et al.[94], represents the cortical labels used to construct the network in (**b**). **d** Additional subcortical labels. A surface rendering of the subcortical labels that were estimated and added to the cortical labels is shown. **e** Trends in behavior over the lifespan. Violin plots show the full distribution of the observations for the Mini-Mental State Exam (MMSE). Error bars represent mean ± 2 standard errors (s.e.). **f** Trends in the connectome over the lifespan. Violin plots show the full distribution of the observations for the highest node degree. Error bars represent mean ± 2 s.e. For panels **e** and **f**, subjects were binned by decades starting at 18 years of age (see ref. [90]).

**Table 1 Open brainlife.io web services and containerized applications implementing the processing pipeline developed for the current work.**

| App name | App purpose | DOI |
|---|---|---|
| AC-PC Alignment of T1 | *Align T1w anatomical images along the AC-PC plane* | bl.app.99 |
| AC-PC Alignment of T2 | *Align T2w anatomical images along the AC-PC plane* | bl.app.116 |
| FreeSurfer (v6.0.0) | *Create surfaces and cortical labels* | bl.app.0 |
| dMRI Preprocessing | *Correct dMRI data of artifacts and align to T1w image* | bl.app.68 |
| RACE-Track (Tractography) | *Create an anatomically constrained whole-brain ensemble tractogram* | bl.app.101 |
| Multi-Atlas Transfer Tool (maTT) | *Map HCP-MMP atlas to individual subjects' brains* | bl.app.23 |
| Network Construction | *Create networks from tractography and cortical labels* | bl.app.121 |

We developed a computationally reproducible data processing pipeline utilizing the cloud computing platform brainlife.io[89]. Each step within the pipeline is briefly described and available for download as a Docker[138] container run via Singularity[139]. Each step of the pipeline is also publicly shared as an App (web service) on the brainlife.io platform. brainlife.io Apps can be freely executed on public or private computing resources. All processed brain imaging data and Apps are accessible in a single record as a brainlife.io publication at[89].

A large subset of behavioral measures varied as a function of age (Fig. 1 and Supplementary Fig. 1). Reaction times increased with age, accuracy decreased, and performance in multiple scales either decreased or increased with age depending on the scale valence (refs. [101,102]; Supplementary Fig. 1a). For example, reaction time increased in the Force Matching Task, but accuracy decreased (Supplementary Fig. 1a), performance in the Mini-Mental State Exam (MMSE) also decreased (Supplementary Fig. 1a; ref. [90]). To approximate these trends we fit a quadratic polynomial to each behavioral measure. We found that 166 of the total 334 behavioral variables increased with age (quadratic term: $0.034 \pm 0.083$ s.d., $R^2 = 0.957 \pm 0.123$ s.d., AICc $= 49.067 \pm 4.977$ s.d.), and the remaining 168 decreased with age (quadratic term: $-0.005 \pm 0.016$ s.d., $R^2 = 0.988 \pm 0.050$ s.d., AICc $= 48.232 \pm 1.079$ s.d.).

To evaluate whether the brain networks generated using RACE-Track showed sensible changes in properties across the lifespan, we estimated the highest node degree (Fig. 1f and we observed similar patterns when evaluating network density and efficiency see Supplementary Fig. 1b). To summarize the effect of age on these three measures, we fit a quadratic polynomial. The highest network node degree was well described by a negative quadratic term; it increased early and decreased later in life (quadratic term: $-0.005 \pm 0.008$ s.d., $R^2 = 0.9996 \pm 0.0004$ s.d., AICc $= 48.007 \pm 0.007$ s.d.; network density and efficiency demonstrated a similar negative quadratic pattern, Supplementary Fig. 1b, right).

**A single mode of covariation relates individual differences in structural networks and behavior with subjects' age.** After replicating the established quadratic trends in brain and network properties and behavioral variables with age, we set out to build a model that would linearly associate all the measurements from the two data domains—brain and behavior. We did this because we wanted to determine whether a linear association well approximated the relationship between brain networks and behaviors. To do so we used canonical correlation analysis (CCA; ref. [83]). CCA finds the linear combination of variables that best associates measures from the two data domains across subjects. In the following analyses, CCA found the best linear combination of 376 brain network properties and 334 behavioral measures (see Fig. 2, "Methods", and Supplementary Fig. 2a, c). To do so, the behavioral measures and networks' node degree estimated in each subject were organized into two matrices ($D_1$ and $D_2$; Supplementary Fig. 2a). Confounding variables such as sex, handedness, height, body weight, heart rate, and blood pressure were regressed out from both $D_1$ and $D_2$ (see Supplementary Fig. 2a, "Methods", and ref. [83]). The eigenvector matrices ($E_1$ and $E_2$) estimated from $D_1$ and $D_2$ via Principal Component Analysis (PCA) were used as inputs to CCA (ref. [83]; Supplementary Fig. 2b, c). CCA found the weights ($a$ and $b$) and canonical factors ($F_1$ and $F_2$) that best approximated $E_1$ and $E_2$ (Supplementary Fig. 2c and d).

A first CCA model ($M_0$) used a large number of PCA eigenvectors (100) as inputs and no cross-validation to evaluate the best possible fitting model to the data[83]. $M_0$ returned multiple large and statistically significant modes of covariation explaining the relationship on the first canonical axis ($CA_1$) between networks and behaviors ($CA_1 = 0.860$–$0.1072$; $p_{1-75} = 0.000$ and $p_{76-85} < 0.01$ bootstrap test). After this exploratory model, a 5-fold cross-validation approach (subjects as a random variable; Supplementary Fig. 2e) was used to further test the association between networks and behavioral variables. A grid-search approach was used to generate 9801 CCA models with different combinations of PCA numbers (spanning from 2 to 100). Each model was cross-validated using 15,000 5-fold throws for a total of 147,015,000 tested models. The first canonical correlation for a

majority of these models was large with a mean $CA_1$ of $0.5468 \pm 0.09$ s.d. (min and max 0.235 and 0.604, respectively). Over 90% of the $CA_1$ values across all models lay above 0.55, this indicated that, with a few exceptions, a majority of the cross-validated models $CA_1$ explained the data reasonably well.

Because of the established association between age and several brain- and behavioral measures (see Fig. 1), we selected the model ($M_1$) with the highest correlation between age and $CA_1$ ($r_{age} = 0.627 \pm 0.02209$). The selected model, $M_1$, had 38 brain and 40 behavior principal components and a significant $CA_1$ of $0.581 \pm 0.0001$ ($p = 0.000$ bootstrap test). Noticeably, $CA_1$ was the only significant mode in $M_1$ (see Fig. 2b, c, see also "Methods" and Supplemental Fig. 2f-h). The remaining modes were either not statistically significant (average $CA_{3-38} = 0.004 \pm 0.014$; $p = 0.442 \pm 0.254$ s.d. bootstrap test) or had no statistically significant loadings ($CA_2$; see Supplemental Fig. 2i, j and next section). $r_{age}$ was computed using a multiway correlation via a general linear model where subjects' age was predicted using $CA_{1,behavior}$, and $CA_{1,network}$ as regressors (see "Methods", Eq. (3)). $r_{age}$ was strong and statistically significant for the first mode as expected given the model selection procedure ($0.627 \pm 0.022$; $p = 0.000$ bootstrap test) and not significant for the rest of the modes (average $r_{age}$ for $CA_{2-38}$ was $0.077 \pm 0.022$; $p = 0.46 \pm 0.029$ bootstrap test).

Two additional analyses we performed to further explore and validate the results. First, $r_{age}$ was also computed for the exploratory mode, $M_0$. In this case, the association was found to be significant for $CA_1$ only ($r_{age} = 0.611 \pm 0.023$; $p = 0.000$ bootstrap test; average $CA_{2-38}$, $r_{age} = 0.063 \pm 0.015$; $p = 0.726 \pm 0.265$ s.d. bootstrap test, see "Methods"). Finally, to validate the consistency of our approach, and the role of age in the association between brain networks and behaviors a new CCA model was specified ($M_2$). $M_2$ was similar to $M_1$ with the exception that the subjects' age was regressed out from both the brain and behavior data before performing the PCA (see "Methods", Eq. (2)). $CA_1$ for $M_2$ was extremely small even though somehow significant ($0.098 \pm 0.0016$; $p = 0.007$ bootstrap test). None of the remaining modes for $M_2$ were significant (average $CA_{2-38}$, $0.0003 \pm 0.017$; $p = 0.520 \pm 0.290$ s.d. bootstrap test; see Supplementary Fig. 2k, l). As expected, the correlation between age and $CA_{1-38}$ for $M_2$ was insignificant (average $r_{age} = 0.075 \pm 0.019$ s.d. and $p = 0.490 \pm 0.300$ s.d. bootstrap test). These results show that the quadratic trends for individual variables across the lifespan (as shown in Fig. 1e, f) in the two data domains (behavior and brain networks) are well coupled by CCA into a single linear trend. In addition, the CCA mode is also strongly associated with the participants' age.

To summarize this main result, we implemented a hypothesis-driven approach to CCA using cross-validation and tested our hypothesis of an association between age and the CCA model. More specifically, we tested the degree to which the participants' age predicted (in cross-validation terms) the variability in the linear combination between hundreds of variables from brain and behavioral measures. The results show that a major portion of the variability in the CCA model is effectively associated with age. This result does not mean that other variables were not associated with the model, indeed they were, as the CCA model is significant and correlation in $CA_1$ is strong.

**Brain network and behavioral components contributing to the CCA.** We further investigated the behavioral domains and brain network nodes that most contributed to $M_1$, and indirectly to the predominant association with the subject's age. To do so, the CCA variable loadings, $L_1$ and $L_2$ were estimated by computing the correlation between each column in $D_1$ and $D_2$ and every column in $F_1$ and $F_2$, respectively (Supplementary Fig. 3a; ref. [83]).

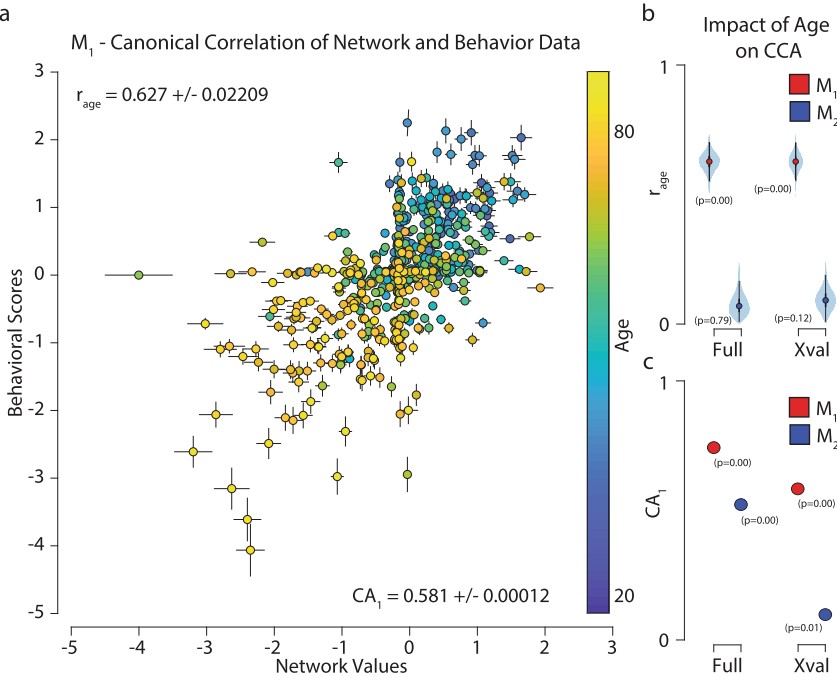

**Fig. 2 Human age explains the majority of the variability in the linear association between brain networks and behavioral variables. a** The first canonical axis (CA$_1$) from the cross-validated model (M$_1$). Each point represents a subject's cross-validated factor score on the brain and behavior axes, respectively. Error bars represent ±3 standard error of the mean (SEM). The color of each point represents the age of the subject. The labeled correlations refer to the correlation of age to the combined canonical axis (r$_{age}$; top) and the correlation between the two canonical axes (CA$_1$; bottom) ±3 SEM. **b** Association (r$_{age}$) between the data in the first canonical axis (CA$_1$) and each participant's age. Violin plots show the full distribution of the observations. Statistical significance estimated by bootstrap. **c** Association between the brain network and behavior data for the first canonical axis (CA$_1$) is shown for both M$_1$ and M$_2$. Statistical significance estimated by bootstrap.

The top 20 L$_1$ and L$_2$ are reported for CA$_1$ (Fig. 3a, b for brain networks and behavior, respectively). The results show that the brain network nodes known to be affected by aging were among the top contributors[34,65]. The inset in Fig. 3a shows the mapping of all the CA$_1$ loadings to the cortical and subcortical surfaces. The visualization of the loadings shows that the distribution of loadings is homogeneously distributed with a few hot-spots, with positive foci in the frontal lobe, the hippocampus, the putamen, and portions of the default-mode network. In the next paragraph, we summarize how the brain area loadings relate to previously reported functional network labels[93]. In parallel, behavioral tasks and scales measuring cognitive and emotional domains also known to be affected during human aging returned meaningful loadings as well[18,66]. These domains consisted of visual recognition, attention, and memory tasks, as well as, reasoning and language comprehension scales. No significant canonical loadings were found for CA$_{2-38}$ (see Supplementary Fig. 3b for CA$_2$ loadings). For completeness, we also tested all the loadings for cross-validated M$_2$. No significant loading was found for this model (see Supplementary Fig. 3c, $p > 0.05$ bootstrap test). In sum, none of the CA in M$_2$ was interpretable, hence the model was uninterpretable.

To summarize the top contributors to M$_1$, word-cloud representations of the variable domains were created (Fig. 3c, d). The word cloud for the brain network loadings was built by assigning the labels from the Glasser atlas to a set of established functional network labels[93], hereafter referred to as Y$_{2011}$. Each of the 376 nodes in our networks was assigned to one of the seven functional networks in Y$_{2011}$: Visual, Somatomotor, Limbic, Ventral attention, Dorsal Attention, Frontoparietal, Default Mode (DMN; see "Methods" for the assignment of the network's nodes to major functional networks in Y$_{2011}$). In addition, the hippocampus, amygdala were kept separated and all the

remaining subcortical structures were combined and reported as subcortical (i.e., pallidum, putamen, accumbens, thalamus, and caudate). This process generated a total of ten words (or clusters). Each word was scaled by the sum of the loadings of all the nodes assigned to the cluster (Fig. 3c, see also "Methods" for a description of how the word clouds were generated from the loadings).

This analysis highlighted the top functional networks and subcortical structures contributing to M$_1$, the one that indirectly also contributed to predicting the subjects' age. The limbic and default-mode networks and the hippocampus returned the strongest contribution (positive and negative, respectively). Other networks, such as the frontoparietal and dorsal attention, as well as the amygdala, provided a strong positive contribution. A similar, word-cloud representation was implemented also for the behavioral variables. Each of the 334 tasks and scales was uniquely assigned to one of the seven behavioral domains described in Fig. 1a. Their loadings were then averaged across all tasks and scales within each domain (Fig. 3d). The word-cloud summary representation for the behavioral loadings shows the variables in the emotional, language, and memory domains contributed the most to M$_1$. Other variables such as social, attention, motor, and clinical scores provided secondary contributions. Next, we evaluated whether the properties of the rich club properties of the brain networks[73,103,104] were associated with the CCA loadings.

**The brain rich-club contribution and canonical correlation.** One of the most reliable findings in network neuroscience is the brain rich-club organization[73,103,104]. Here, we were interested in estimating whether there was a relation between the top contributing brain regions in CA$_1$ and the regions participating in the rich-club organization of the brain. More specifically, we

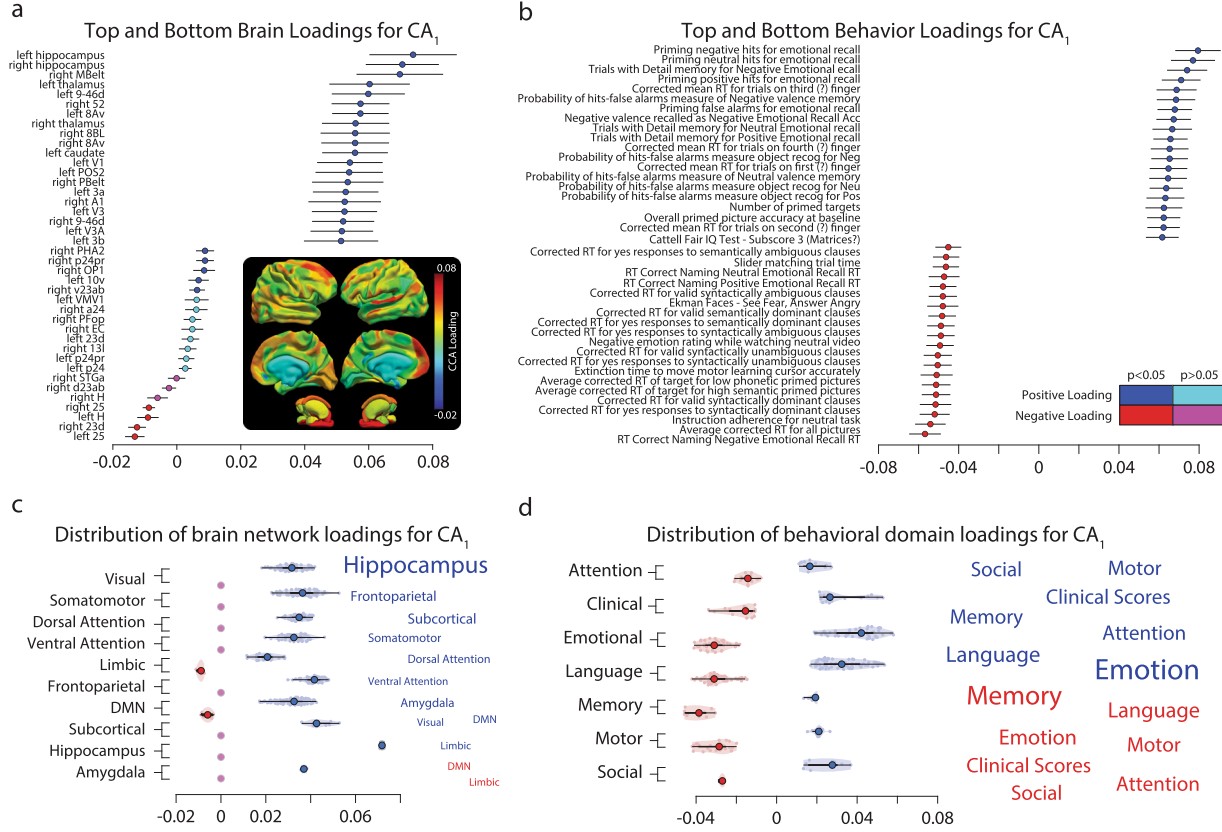

**Fig. 3 Top brain network nodes and behavioral variables contributing to the canonical correlation analysis. a** The highest and lowest contributing cortical nodes within the network. Top and bottom 20 $CA_1$ brain regions sorted by loading magnitude. Positive and significant loadings are shown in blue, negative in red. Non-significant loadings are shown in cyan or magenta. The anatomical inset visualizes the $CA_1$ loadings projected on the cortical and subcortical regions used to build the networks. **b** The highest and lowest contributing behavioral scores within the domain. The top and bottom 20 $CA_1$ behavioral scores sorted by loading magnitude (same conventions as in **a**). **c** Left: Distribution of $CA_1$ loading by the major functional brain network as described by $Y_{2011}$[93]. Right: Word cloud of the functional network name scaled by contribution to $CA_1$ (the larger the font the greater the contribution). **d** Left: Distribution of $CA_1$ loading by behavioral domain. Right: Word cloud the behavioral domains scaled by contribution to $CA_1$. Symbols report mean ± 2 standard errors (s.e.) estimated across 10,000 cross-validation throws of $M_1$. Violin plots report the full underlying distributions of the data, when possible.

evaluated the extent to which the top $CA_1$ brain loadings (i.e., Fig. 3a) mapped onto the rich-club core or periphery. To do so, the mean network across subjects, $N_\mu$, was generated by averaging streamline density for each edge (edges not appearing in at least 50% of the subjects were set to $0$[91]). The number of nodes' participating in the rich club depends on the brain parcellation used. In our study, the rich-club core was defined as the top 15% highest-degree nodes in $N_\mu$. This proportion of rich-club nodes was previously reported by van den Heuvel and Sporns[103] (see also "Methods" for more details). Fifty-four regions from the HCP-MMP (v1.0) parcellation and subcortical labels were assigned to the rich-club core. The remaining 376 regions were assigned to the rich-club periphery (Fig. 4a blue and gray, respectively). We found that this procedure successfully assigned regions to the rich club nominally matched the regions reported in the literature (e.g., superior parietal, precuneus, superior frontal cortex, putamen, hippocampus, and thalamus; Fig. 4a; see also ref. [103] for a comparison).

After defining the rich-club core and periphery, we correlated each node's loading on $CA_1$ with the normalized node degree used to estimate the rich-club participation. The $CA_1$ loadings for regions within the rich-club core or periphery were averaged together. On average, the loadings were higher for the core than the periphery ($0.045 \pm 0.0096$ s.e., $p = 0.00$ and $0.032 \pm 0.0071$ s.e., $p = 0.00$, respectively). This shows a trend for higher $CA_1$

loadings within the rich club, even though no significant difference was found in the mean loadings between core and periphery (Fig. 4b; $p = 0.722$ bootstrap test). Supplementary Fig. 4 shows the rich-club participation coefficient and $CA_1$ loadings for each brain region color-coded by their participation to the rich club or periphery.

We also tested whether the measure used to define the rich club (the node degree of the average network) was significantly correlated with the brain variable loadings in the network axis of $CA_1$ (for reference the values shown in Fig. 3a). We report two findings. First, a medium-strong correlation was found if the rich-club organization was disregarded and the correlation computed across all brain regions ($r = 0.642 \pm 0.034$ s.e., Spearman rank $r$; $p = 0.00$ bootstrap test). The correlation was weaker when we considered only the 54 regions within the rich-club core ($r = 0.26 \pm 0.127$ s.e., $p = 0.0283$), and larger when we considered the 322 regions in the rich-club periphery ($r = 0.56 \pm 0.041$ s.e., $p = 0.00$). Overall, the results show an interesting trend with higher $CA_1$ loadings for regions within the rich club compared to regions in the rich-club periphery (a statistically non-significant trend) and a statistically significant correlation between the node degree in the rich-club periphery. The results can be interpreted as indicating a heterogeneous contribution of regions within and outside the rich-club in the association between brain and behavior and in predicting age.

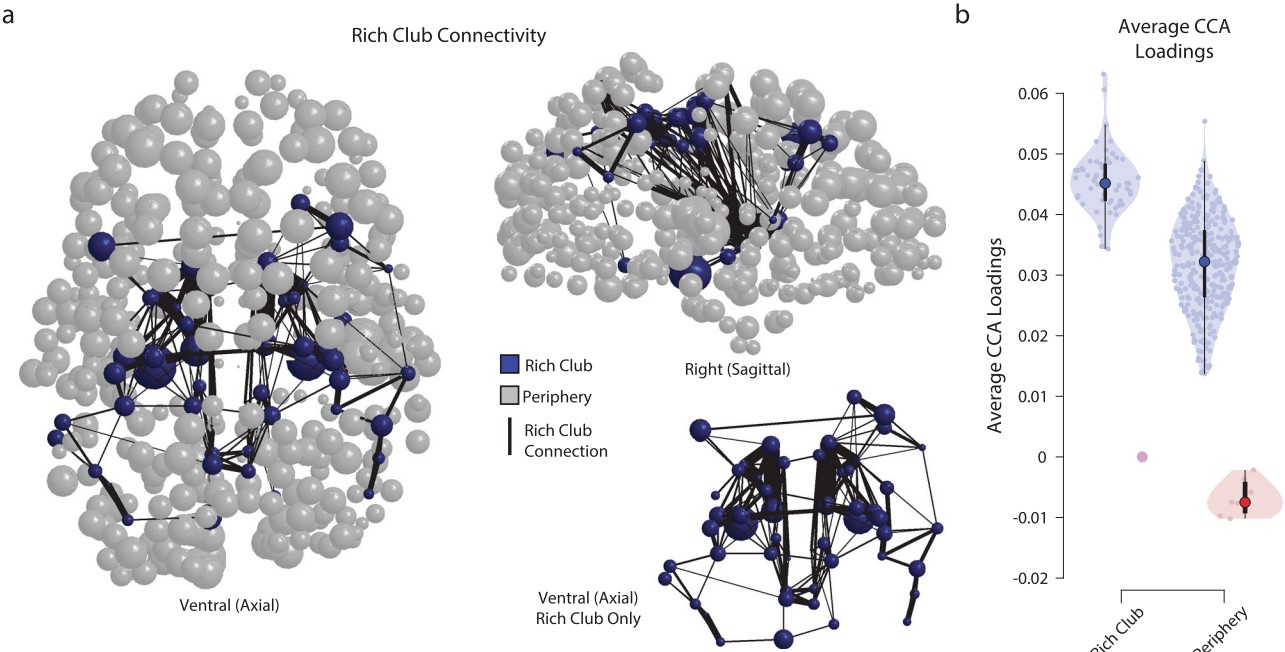

**Fig. 4 Relationship between the brain connectome rich-club and the contribution to the CCA. a** A ball-stick representation of the rich club scaled by CCA loadings. Axial and sagittal views of the relation between rich club and CCA loadings. Each brain region is displayed as a sphere. Blue spheres are part of the core rich-club. Gray spheres are part of the periphery of the rich-club. The diameter of each sphere was scaled by cross-validated $CA_1$ loading in $M_1$. The black lines show connections between rich-club nodes. The thicker the line, the higher the average streamline density. The right-hand bottom ball-stick representation shows an axial view of the rich-club nodes and connections isolated from the periphery. **b** The average $CA_1$ loading across rich-club core and periphery. Positive and negative $CA_1$ loadings for rich-club core and the periphery (violin plots report the full underlying distributions of the parameters from $CA_1$).

**Describing the multivariate relationships between networks and behaviors using representational similarity analysis.** Our overarching goal was to describe a multivariate fingerprint of the associations between brain networks and behavioral domains. Most previous analyses looked at pairs of associations to characterize the changes in individual cognitive tasks and brain systems as a result of aging[21,51,65]. This approach focussed on understanding the relationships between individual cognitive functions and brain systems across the lifespan[66–69]. As a result a global fingerprint of the multivariate changes in the brain network connectivity, behavior, and cognition associated with human aging, have not been described. We used representational similarity analysis (RSA; refs. [92,105]), to develop a many-to-many map between behavioral domains and brain network systems.

Our analyses in the previous sections focussed on $M_1$ to establish the relationships between brain and behavior across the lifespan and how fundamental brain network properties contribute to such relationships. Hereafter, we focussed on the multivariate relationships between brain network and behavior domains. To do so, we derived an approach that used the CCA variables loadings ($L_1$ and $L_2$, Supplementary Fig. 5a) as inputs to RSA. An RSA typically quantifies the dissimilarity between variables from two data domains; for example and more commonly, brain function and behavioral performance[92,105]. In our application instead of using the data from the two domains directly, we used the variable loadings estimated for each canonical axis in $M_1$. More specifically, every variable loading in $M_1$ (376 network and 334 behavior variables) was correlated with the loadings of all the remaining variables, and dissimilarity was then computed using Eq. (4). This process generated a square, symmetric RSA matrix, $S_1$, of size $710 \times 710$.

This approach leverages the inherent structure of the data captured by the CCA model to describe how the loadings of each

brain and behavior variable are associated amongst themselves. The assumptions of this analysis are that the CCA variables loadings contain a fingerprint of the multivariate associations of the brain and behavioral variables. The approach allowed us to model (1) the brain network-to-network similarity, (2) the behavior-to-behavior similarity, as well as (3) the brain network-to-behavior similarity. $S_1$ was summarized by averaging the RSA values from the individual brain network nodes and behavioral variables within the 10 brain networks of $Y_{2011}$ and 7 behavioral domains (as described in the previous section and in "Methods"). This reduced the dimensionality of the symmetric dissimilarity matrix from 251,695 (upper diagonal of the original $710 \times 710$ matrix) to 136 (upper diagonal of the $17 \times 17$ matrix; see Supplementary Fig. 5c). $S_1$ can be divided into three primary regions; brain network-to-network, behavior-to-behavior, and behavior-to-network. Because the interest here was to capture the unique relationship between brain and behavioral variables, *btn* was the focus of all the following analyses (Supplementary Fig. 5d). The results were visualized using a modified chord plot that allowed us to show how multiple associations between brain and behavior load onto $M_1$ simultaneously (Supplementary Fig. 5e; see "Methods"). The chord plot was generated by thresholding the RSA values in behavior-to-network the top quartile. Two aspects of the chord plots shown in Supplementary Fig. 5e are of interest. First, the number of domains (or networks) that each network (or domain) contributed to is described by the size of the peripheral segments for each network and domain. The larger the size of the segments, the more contributions of a network (domain) to the various other domains (networks). Second, the associations between each functional network and behavioral domains are described by individual chords (if a chord exists two domains or networks interacted in the CCA).

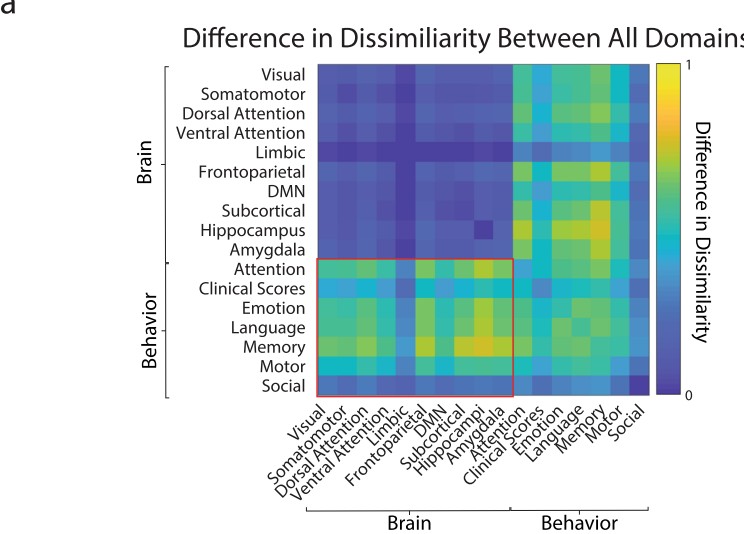

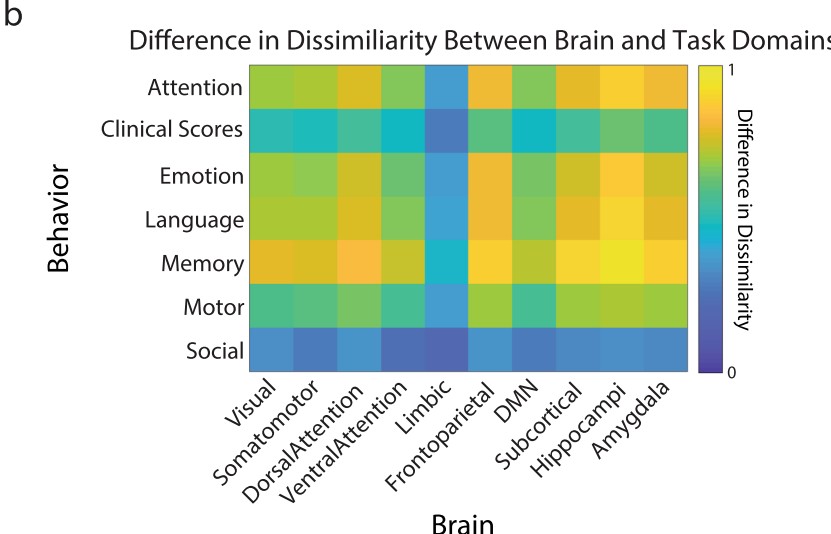

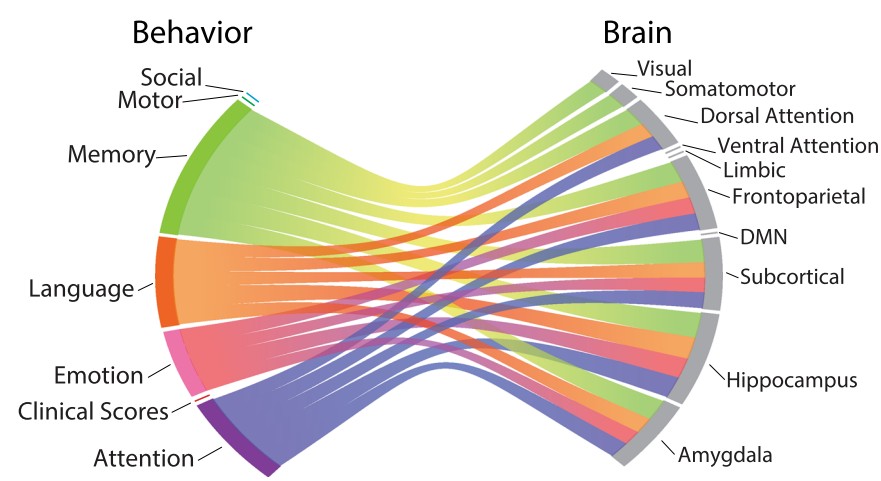

**The difference between RSAs shows multivariate brain-behavior associations the variability accounted for by age.** Hereafter, we wanted to further emphasize the contribution of the individuals' age to the RSA. To do so, we used the second CCA model, $M_2$, in which participants' age was subtracted out (deconfounded). An RSA model was set up using the variable loadings in $M_2$ to create $S_2$. The impact of age was then isolated by subtracting $S_2$ from $S_1$ to compute $S_d$ (Fig. 5). $S_d$ describes the multivariate (many-to-many) associations between the networks and behavioral domains after the variables' associations independent of age have been removed. In other words, the matrix, $S_d$, evidences the dissimilarity values that are related to age but not to the rest of the variables in the model because any common dissimilarity was subtracted out. Because the interest

**Fig. 5 Difference in representational similarity analysis (RSA) shows many-to-many associations between the modules from the brain networks and behavioral domains. a** Differences in RSA ($S_d$) between two CCA models ($M_1$ and $M_2$). The values in the matrix $S_d$ represent the difference between $S_1$ and $S_2$. The higher the RSA value, the higher the impact of age. The red rectangle represents the subset of brain-behavior interactions emphasized in (**b**) (referred to as *btn* in the main text). **b** Subset of the multivariate interactions between brain networks and behavioral domains. The data in panel **b** shows a subset in panel (**a**) emphasized by the red rectangle, also referred to as *btn* in the main text. This subset shows the multimodal dissimilarity between brain and behavior domains across the lifespan. **c** Top quartile of the differences in dissimilarity between brain networks and behavioral domains. This modified chord plot represents the top interactions between the brain networks and behavioral domains in *btn*. Associations below the 75th percentile were removed from the visualization. Two aspects of the chord plots can be appreciated. First, the number of domains or networks that each network or domain contributes to is shown by the size of the peripheral wedged-segments for each network and domain. The larger the size of the segments, the more contribution of a network (domain) to the various other domains (networks). Second, the associations between each functional network and behavioral domains are described by individual chords (if a chord exists two domains or networks interacted in the CCA).

was to capture the unique relationship between brain and behavioral variables, the following description focuses on the cross-data-domain interactions (Fig. 5b). A chord plot was used to describe the many-to-many relationships (Fig. 5c).

The results of this RSA analysis were consistent with previous reports[34,38–40]. More specifically, results showed that the behavioral domains most affected across the lifespan were memory, language, emotion, and attention (i.e., the behavioral domains passing the top 25% cut-off threshold; Fig. 5c). Going beyond previous reports, the RSA results showed that the top behavioral domains were associated not just with a specific brain network, but with multiple of them. More specifically, memory concurrently associated through the lifespan with the hippocampus (top association), the subcortical structures, the frontoparietal network, amygdala, dorsal attention network, the visual, somatomotor networks but the ventral attention, default mode, and limbic networks (see Supplementary Table 1). Emotion was associated through the lifespan with changes in the subcortical structures but also by the frontoparietal network, the hippocampus, and the amygdala. Similarly, attention was affected by the dorsal attention and frontoparietal network but also by the amygdala, hippocampus, and subcortical structures. Language and memory show a similar pattern, with moderately high associations between many brain networks with notably strong connections to the hippocampus. Whereas some of the previous work has been devoted to characterizing individual cognitive tasks and brain systems in relation to aging[21,51,65], the current analysis showed that multiple brain networks meaningfully contribute to the variation in behavioral performance and cognition across the lifespan. The many-to-many relationships between brain networks and behaviors reported here are unexplored but critical to understanding the process of aging in normal and diseased populations.

## Discussion

Brain aging and disease have profound effects on society. Because of the increasing expected lifespan of the world population—many of us are living longer—it is increasingly important to understand how we can age healthily. Costs associated with an aging population are only expected to increase over the next few years, given the increase in life expectancy (1–5). In Europe, estimates reach up to 200 million individuals affected yearly, directly or via caregivers' networks. Such impact accounts for up to 800 billion euros in annual costs. For example, brain disease is likely to affect up to 25% of the European population and nearly 38% of the remaining global population. These numbers make understanding the brain and its relation to behavior paramount to the well-being of society across the globe. Critical to this goal, is to increase our understanding of the normal variability across individuals in the large population is fundamental to improve the prodromal identification of individuals at risk versus individuals subject to normal aging (6–10). To address the need to capture

human variability and aging at least three aspects of scientific inquiry will need to be addressed. First, more high-quality data will need to be connected to reach a population-level understanding of the human brain and behavior. Second, advanced computational approaches will need to be developed that can actually exploit the value of the data at the right scale. Finally, infrastructure will need to be in place to support large-scale computational analysis of thousands of individuals.

To understand aging we need large-scale, population-level studies. There is an increasing need to better understand the trajectory of healthy aging as a large portion of the world's population rapidly progresses into old age. The need to understand not only the expected course of senescence, but the interaction of brain structure and behavior will become increasingly important to understand the myriad of conditions that can impede independence later in life. Early open projects of this nature, such as the Alzheimer's Disease Neuroimaging Initiative (ADNI), have required large publicly available tools to facilitate the larger number of researchers who have analyzed factors related to ones in this dataset[15,16,106–109]. More recent, large-scale population projects such as the Human Connectome Project, the ENIGMA Consortium, and the UK Biobank[60,110,111] have collected, organized, and openly distributed data to allow the implementation of data-driven neuroscience methods, mine novel findings, and build normative models of brain structure and function[94]. These projects are already pursuing similar methods, and the presented analysis could easily be applied and extended to these datasets[38,60,71,83,110].

To fully exploit large-scale datasets advances in both computational methods and data-analysis infrastructure are needed. We described a multivariate analysis of a multimodal dataset[90]. Multiple measures of brain structural connectivity and behavior were combined to estimate the impact of human age on multiple brain systems and behavioral domains. A model testing framework was developed using cross-validation to demonstrate a strong linear association between brain connectivity and behaviors. The majority of the variance in the linear association between connectivity and behavior was accounted for by the first mode of the model. This model was selected to maximize the correlation with subject age. Data derivatives (https://doi.org/10.25663/brainlife.pub.21), code (https://github.com/bcmcpher/cca_aging), and reproducible cloud services (see Table 1) developed for the present work are made available for reuse by the wider scientific community. Finally, the RSA analysis showed a series of global patterns of association between the model, selected to maximize the correlation with the subjects' age, and that resemble findings reported when individual variables are measured individually.

In previous studies, most commonly, a few behavioral or psychological variables have been singled out and related to the changes in brain networks to capture the aging process across the lifespan[21,51,65]. Describing how many of these cognitive and

behavioral variables change across the lifespan has been a challenge[62,73–79]. For example, memory has been related to hippocampal volume and reaction time[43,44,78]. However, these targeted studies run the risk to miss that the hippocampus also has significant contributions to emotion processing and guiding attention[28,34]. Here, we proposed an approach that allowed us to represent the changes in multiple brain network systems and behavioral domains simultaneously. Our results provided a way of quantifying the multivariate relationships to understand how these variables simultaneously change across the lifespan. The results show the changes in behavioral and psychological domains are interwoven across the lifespan. We demonstrate that individual behavioral domains are effectively associated with changes to multiple brain network systems. So whereas focussing on individual variables is helpful sometimes, it is not simply the case that individual behavioral variables affect individual brain areas or network systems. Opposite to that, the picture painted by our results show that many cognitive and emotional domains simultaneously vary across the lifespan with the changes in multiple brain networks. In sum, our results show major changes to behaviors and network structure that cannot be reduced to changes to any individual cognitive constructs. Instead, multiple cognitive and behavioral domains covary in ensemble with global changes to network structure. Future studies will be necessary to better describe the way the brain and behavior relationships reported here interact and how that drives aging in health and disease.

We demonstrate that brain and behavior show multiple global changes across the human lifespans that go well beyond one-to-one correlations between a single brain system or behavioral task. Several previous studies inspired this work, providing ideas for the contributions produced in this study. Early work[83] integrated functional brain networks and phenotypic data from a large dataset of healthy young adults[110]. A strong association between brain function and behavior was reported using CCA. Yet, age was not a significant result in the analysis as it was in ours, most likely due to the narrow age range in the Human Connectome Project sample used in that work compared to the CAN sample. Another relevant study used the CAN dataset, CCA, and generative models of brain functional activity and was first in reporting the effect of age on the association between cognitive performance and brain function[38]). The study focused on three functional networks[112,113] and six composite measures of behavior and showed that neuronal, not vasculature-related effects in resting-state networks are associated with age. A third relevant study was first in using brain structure (a multimodal combination of diffusion-measures and cortical partial volume estimates) to show an association with various demographics, including age[84]. The current results go beyond previous work by (1) thoroughly cross-validating the CCA, (2) testing how well a hold-out variable is predicted by the primary canonical axis, and (3) using an RSA on the variable loadings from the cross-validated model to recover relationships between the variables themselves. These improvements provide a way for future work to glean more insight from the increasingly large and multimodal samples of variables utilized in modern neuroscience. Furthermore, we created open services shared on brainlife.io to allow other researchers to freely process new data utilizing our pipeline.

As neuroscience shifts toward data-driven approaches, data-fusion methods[114–116] such as CCA are likely to increase in popularity. Yet, the CCA approach used here is limited to two domains at a time. Other methods allow fusion across a higher number of data domains. For example, CCA has been generalized to higher-order models beyond 2 domains, similar to variations of Independent Component Analysis (joint- or linked independent components analysis[117–119]. Imaging genetics is one of the fastest growing fields that are capitalizing the most on methods such as CCA especially requiring to combine multiple data domains[6,16,106,111,120,121]. These studies typically look at thousands of genetic markers from a single blood assay, similar to how a single MRI imaging session can be used to generate thousands of measures of brain structure and function. The benefit of methods such as CCA is due to the ability to model datasets with large numbers of variables from different domains (e.g., blood assays, behavior, genetics, and neuroimaging). Yet, in addition to variations of CCA, independent components analyses have also been proposed to map across multiple variable domains, though Partial Least Squares (PLS; refs. [82,122,123]) is perhaps the most common approach that allows mapping the combination of multiple data modalities into a single space[124,125]. Furthermore, the current work is also limited to linear interactions between data domains. Future explorations of the multivariate interactions might contribute additional insights[126,127]. Nonlinear interactions estimated through kernel embedding[128] or structural equation modeling[129] are particularly promising, but have not yet been applied to datasets similar to the one used here. Critically, one recent report has sparked interest by suggesting that adding nonlinear interactions to data modeling may provide negligible improvement over linear models when comparing brain-behavior interactions[130]. In selecting the best models (linear or nonlinear) might also depend on data preprocessing[131,132]. The effectiveness of data preprocessing may vary depending on the characteristics of the data domains. Future work will be required to determine what data domains will require linear or nonlinear modeling approaches.

As neuroscience moves to population-level neuroscience, reproducible approaches to data management and analysis need to be embraced. Sharing data products is essential to implementing transparent, replicable, and reproducible brain research (23). Critical to the success of the next-generation large-scale neuroscience methods will be to embrace new technology to ensure results reproducibility. This will require embracing methods for open science, data, and computational standards as well as modern computing infrastructure to lower the barriers of entry to proficient large-scale data methods (24–26). Our study follows up on the most recent trends in terms of large-scale datasets and computational approaches. But also the study embraces the most recent technology to support scientific reproducibility to clarify human aging. We used the recently developed, community-developed, and publicly-funded cloud computing platforms, brainlife.io. brainlife.io allows processing large amounts of data by tracking the provenance of each dataset and by linking the data-object on the platform with the reproducible web services used to generate the data. brainlife.io addresses precisely the needs for replicability and reproducibility highlighted by the recent report by the U.S. National Academies (23). Indeed, to implement our study we developed more than ten new data analyses applications that are now available on brainlife.io for other investigators to reuse for new research or to replicate our results.

## Methods

**Data source**. The behavior and neuroimaging data were accessed from the publicly distributed Cambridge center for Ageing and Neuroscience (CAN) dataset[90]. This large dataset provides cross-sectional data on 652 individuals, who provided informed consent. The acquisition of this dataset was approved by the Cambridge University Ethics committee and its distribution complies with the Helsinki Declaration. Of this total 623 had imaging data of interest to the current study (diffusion-weighted MRI and T1-weighted MRI). Out of a total of 623 individuals, 594 neuroimaging datasets were successfully processed and included in this present study. Quality control issues on the processed T1w data (FreeSurfer did not successfully segment the brain) motivated the exclusion of 26 subjects. Supplementary Fig. 2a shows the dimensionality of the data ($n = 594$).

**Behavioral data**. Participants responded to a series of screening and demographic questionnaires and performed behavioral tasks and clinical tests[90]. See ref. [90] for details on the tasks, tests, and questionnaires. A total of 1708 measurements per individual were acquired by the CAN consortium. Only behavior scores normalized by the CAN consortium were utilized for the current study, as a result, a total of 388 CAN-normalized scores were utilized for further analysis. All these measurements were initially subdivided into five of the primary behavioral domains: attention, memory, language, emotion, and motor[90]. For the current study, in addition to the five original behavioral domains, test scores collected by the CAN consortium measuring social engagement and clinical scores were also utilized and collected into a Social and Clinical behavioral domain for a total of seven domains (Fig. 1a). The social engagement measures consisted of 10 questions related to the frequency and mode of socialization of the individuals. Six clinical questionnaires were collected into the clinical domain: the MMSE, ACE-R, Wechsler Memory test, Spot the Word, the Cambridge 10MQ, and the PSQI Sleep Index[90,93].

### Neuroimaging data preprocessing

*Anatomical MRI data preprocessing.* A T1w and T2w anatomical scans were acquired for each individual. Both scans have a 1 mm isotropic resolution (T1w: 3D MPRAGE, TR = 2250 ms, TE = 2.99 ms, TI = 900 ms; FA = 9 deg; FOV = 256 × 240 × 192 mm; GRAPPA = 2; TA = 4 min 32 s. T2w: 3D SPACE, TR = 2800 ms, TE = 408 ms, TI = 900 ms; FOV = 256 × 256 × 192 mm; GRAPPA = 2; TA = 4 mins 30 s). Both images were reoriented and aligned to the AC-PC plane with a 6 degree of freedom rigid alignment to the MNI template based on the Human Connectome Project (HCP) procedure (bl.app.15, T2 reorientation; brainlife.app.114, T1 alignment; bl.app.99, T2 alignment; brainlife.app.116). These AC-PC-aligned images were passed to Freesurfer 6.0[133], which reconstructs the white-matter and pial surface based on the T1 image, using the T2 image for additional information to better estimate the surfaces (FreeSurfer; bl.app.0). The FreeSurfer output was used by the multi-atlas transfer tool (maTT; bl.app.23) to align individual subjects T1w anatomy files to the multimodal cortical brain atlas (HCP-MMP v1.0 Atlas[94]. The labels from the HCP-MMP atlas were used to build networks, see below.

*Diffusion-weighted MRI data preprocessing.* The diffusion-weighted MRI data (dMRI) contained a total of 60 directions collected across 2 b-value gradients (2D twice-refocused SE EPI, TR = 9100 ms, TE = 104 ms, TI = 900 ms; FOV = 192 × 192 mm; 66 axial slices, 2 mm isotropic; B0 = 0, 1000/2000 s/mm$^2$, 30 unique directions per shell; TA = 10 min 2 s. Readout time 0.0684; echo spacing = 0.72 ms, EPI factor = 96). Data were processed using MRtrix3 (bl.app.68; refs. [96,134]). The following are the steps of the preprocessing procedure. The diffusion gradient alignment to the data was checked using a simple tracking procedure to find the orientation that produces the longest streamlines. After that, a PCA denoising procedure was performed to remove scanner noise not associated with the diffusion signal. Gibbs ringing, Eddy currents, inhomogeneity, and motion artifacts correction was performed. After all previous steps, an additional Rician-denoising was performed. The mean B0 image across (2 repeats) was extracted and utilized to register the dMRI to the AC-PC aligned T1-weighted image using Boundary Based Registration (BBR, ref. [135]). The diffusion-weighting gradients were rotated to account for motion correction and AC-PC alignment. Finally, the diffusion-weighted image was resampled to a 1 mm isotropic.

*Tractography generation.* We developed an automated cloud computing service that we call Reproducible Anatomically Constrained Ensemble Tracking (RACE-Track; bl.app.101). RACE-Track combines Anatomically constrained tracking[98] with ensemble methods[97] and is containerized for public, reproducible access on the open cloud computing platform brainlife.io[89]. The preprocessed dMRI data was used to generate whole-brain tractography using RACE-Trac. Specifically, a constrained spherical deconvolution (CSD) response function was estimated for white matter, gray matter, and cerebrospinal fluid (CSF) as described in refs. [134,136]. Each respective response function was then used to estimate Fiber Orientation Distribution (FOD) maps for each tissue type. Following the ET approach, we fit the CSD model using multiple $L_{max}$ (2, 4, and 6), this procedure created three FOD maps one per $L_{max}$. The FOD maps were then passed to the iFOD2 tracking algorithm as described in ref. [98]. The T1w image was used to separate the brain tissue into its five types: (1) corticospinal fluid, (2) white matter, (3) gray matter, (4) subcortical gray matter, and (5) pathological tissue. The image probability maps generated for each tissue were used for initiating and stopping tractography. More specifically, the gray-matter–white-matter interface was used as a seed mask for the streamlines This approach has been demonstrated to provide a more complete coverage of tractography terminations on the cortex and subcortical gray-matter structures[98]. Furthermore, the whole tractography procedure was repeated for both deterministic and probabilistic tracking, across the three $L_{max}$ and five different maximum angles of curvature (5, 10, 20, 40, and 80°). A total of 600,000 streamlines were generated in each individual using this tracking procedure.

*Construction of structural brain networks.* Whole-brain structural brain networks were built utilizing the multimodal cortical brain atlas (HCP-MMP v1.0; ref. [94]). The HCP-MMP atlas was aligned to individual subjects' brains with maTT (bl. app.23). The parcels of the HCP-MMP were used as nodes for the brain networks.

The edges of the networks were defined as the density of connections between two parcels. Connection density was computed as the number of streamlines terminating in the two parcels divided by total number of voxels in the parcels[100]:

$$C_d = (2 \cdot s_{i,j})/(n_i + n_j) \quad (1)$$

Where $C_d$ is the estimated streamline density for a connection (network edge), $s_{ij}$ is the number of streamlines terminating in both regions, $n_i$ and $n_j$ are the number of voxels in each of the two brain regions. Network matrices were created using Eq. (1) using a reproducible algorithm implemented on brainlife.io (bl.app.121). After that, the network matrices were thresholded by removing connections not present in at least half of the participants and then computing node degree for each individual network matrix using the Brain Connectivity Toolbox[91].

### Canonical correlation analysis and data preprocessing

*Behavior features.* The total of 388 behavioral variables ($m_1$; Supplementary Fig. 2a) were extracted from the CAN project. Data were prepared for modeling by applying a z-score transformation to each measurement across subjects[83]. Furthermore, variables with extreme outliers (more than three standard deviations from the population mean) or absent in at least half of subjects were eliminated from further analysis. Fifty-four of the behavioral variables were eliminated via normalization, bringing the total number of behavioral data utilized 388 to 334. These features were organized into a matrix ($D_1$) composed of all behavioral (594 subjects, $n$, and 334 behavioral variables, $m_1$). See Supplementary Fig. 2a and Eq. (1).

*Network neuroscience features.* Node degree was estimated using the streamline density networks generated described above for each subject[91]. This estimation procedure resulted in 376 node-degree measures per subject ($m_2$). The upper diagonal of each network was unwinded to create a vector of features 1 × 70,500 in size[83]. These features were organized into a matrix ($D_2$) composed of all brain data (594 subjects, $n$, and 70,500 node-degree estimates, $m_2$). See Supplementary Fig. 2a and Eq. (1).

*Data normalization.* Both the brain and behavior datasets were normalized through the same process. First, the datasets were normalized by removing the mean of each feature across subjects and dividing by the standard deviation across subjects. Next, nuisance variables were regressed from each dataset to remove their influence on the data. These variables include a subjects' height, weight, gender, heart rate, and systolic and diastolic blood pressure. Behavior data were also rotated to the nearest positive definite pairwise covariance matrix to better account for correlation between variables[83]. This step was not required and not performed on the brain network data. Principal component analysis (PCA) was performed on $D_1$ and $D_2$ separately to reduce the number of dimensions in the data. PCA generated two eigenvector matrices, $E_1$ and $E_2$, for $D_1$ and $D_2$, respectively (see Supplementary Fig. 2b and Eq. (1)). Because the choice of number of principal components (PCs) is arbitrary, we explored a wide range of PCs (2–100) and selected the PC number for $D_1$ and $D_2$ that best predicted the individual subjects' age (see below for more details).

*Canonical correlation analysis (CCA).* We used $E_1$ and $E_2$ to perform canonical correlation analysis (see Supplementary Fig. 2c). The CCA algorithm estimates the linear combination of variables from each dataset that maximizes the correlation between the datasets

$$\rho = \max_{a,b}[Corr(a^T D_1, b^T D_2)]\%0 \quad (2)$$

where $\rho$ is the correlation between the datasets being maximized, $D_1$ and $D_2$ are the normalized datasets, and $a$ and $b$ are the weighting matrices that linearly combine the data variables into the canonical scores. The CCA analysis returns two matrices, $F_1$ and $F_2$, containing the canonical factor scores for each subject to each of the estimated canonical factors. The factor scores describe how much each subject contributes to each of the latent factors. These factor scores are used plotting the canonical correlation axes, or canonical axes (CA, Fig. 2, Supplementary Fig. 2, red). We use the combination of the first factor scores from each dataset to estimate the correlation with age.

To better interpret the contribution of individual variables within the CCA factors, variable loadings are reconstructed from the normalized data and the canonical factors. The loading values, $L_1$ and $L_2$ for brain and behavior, respectively, describe the strength of the variable's contribution to the axis. The loading matrices' dimensions are the number of variables in the dataset by the number of estimated canonical factors. Each variable has a loading for every factor. The loadings are recovered for a variable by correlating the original normalized variables for every subject with the corresponding factor scores. This process is repeated and every variable is correlated with a factor to create the variable loadings for a canonical axis. By correlating the normalized variable observations (each column of matrix $D$) to each factor score (each column of matrix $F$) (Supplementary Fig. 2) for every canonical factor estimated in both datasets, we generate canonical loadings ($L$) for every variable and every factor. For example, a specific behavioral score could be highly correlated with a canonical factor, indicating that the behavior is highly predictive of changes in this factor. When this

is done for every axis, a complete matrix $L$ of variable loadings is recovered. The highest variable loadings for a canonical axis indicate the variables that contribute most to the positive end of the canonical factor, while the lowest variable loadings indicate the variables that contribute most to the negative end of the canonical factor. This is reported in Fig. 3. The variable loadings are the typical way of inspecting how variables within a CCA analysis interact.

*Correlation with age.* To determine the association of a canonical axis with age, we used a multiway correlation with the first canonical factors to determine an $r$ coefficient between the canonical axis and age. Age was predicted by

$$\text{age} = \text{CA}_{\{1,\text{network}\}} w_1 + \text{CA}_{\{1,\text{behavior}\}} w_2 + b \qquad (3)$$

where age is the age of each subject, $\text{CA}_{1,\text{network}}$ is every subjects score in the first column of the brain factors ($F_1$), $\text{CA}_{1,\text{behavior}}$ is every subjects score in the first column of the behavior factors ($F_2$), $w_1$ and $w_2$ are the beta estimates of the model, and $b$ is the intercept. This is an ordinary least squares linear regression using the canonical scores of the behavior and brain from the first factor predicting age. To determine the correlation between the factors and age, we take the square root of the $R^2$ from the fit model to estimate the correlation coefficient between the combination of factors and age. This is equivalent to the correlation with age along the canonical axis of the estimated factors. To determine the significance of the correlation of age to the canonical axis, a bootstrap test of 10,000 permutations was performed by randomly sorting the rows with replacement of $\text{CA}_{1,\text{network}}$ and $\text{CA}_{1,\text{behavior}}$ to determine a null prediction. This allowed us to determine if the observed correlation with age was significantly different from zero. To estimate the standard error of the estimate, a separate 10,000 permutation bootstrap was estimated, sampling with replacement of both factors and age. The resampled estimates of age were used to create a rempled mean and standard error of the correlation between the canonical factors and age.

*Parameter tuning.* The goal of this analysis is to maximize the correlation with age. However, the principal components analysis performed as part of the preparation of the data for the CCA will combine variables differently depending on the number of components requested (Supplementary Fig. 2f, g, h). Further, the number of PCA components indicates the number of canonical factors that can be estimated. Typically, a number such as 20 or 100 is selected by the researcher. However, there is reason to believe that the number of requested components will impact the final correlation with age (Supplementary Fig. 2f, g, h). To test this we tuned the number of PCA components using a grid search, selecting the number of principal components based on the strength of the resulting correlation with age. This hyperparameter sweep found the correlation with age on the primary canonical factors for every combination of PCA components between 2 and 100 on the input datasets. See Supplementary Fig. 2e, d for the results from the search space. We selected 38 principal components for the brain axis and 40 principal components for the behavior axis because this combination of principal components resulted in CCA factors with the highest correlation with age. Nearby parameter spaces had similar levels of correlation with age. Importantly, the most common combinations of parameters, $[20 \times 20]$ and $[100 \times 100]$, had a significantly worse correlation with age than the model with the optimal parameters (Supplementary Fig. 2f, g, h). This is further supported by inspecting the variable loadings of the models that performed worse than the tuned solution. The variable loadings in the suboptimal models had different combinations of variables as the strongest and weakest loadings. This difference indicates that the CCA used a different combination of variables to form the canonical factors in these models. These different factors are less effective at predicting age. This indicates that the selection of the number of principal components for this analysis is important in tuning an accurate model.

*Model selection.* All factors and loadings were cross-validated using a repeated 5-fold process to better ensure the generalizability of this model to the larger population (Supplementary Fig. 2e). A repeated k-fold process is conceptually similar to a typical k-fold approach. A standard k-fold approach models data by training the parameters on a majority portion of the subjects and then applying the fit parameters to a hold-out set of subjects to observe the generalizability of the trained parameters to new samples from the same population. The two key differences between the repeated k-fold process and a standard k-fold process is that (1) in a repeated k-fold process all observations end up with a held-out estimate and (2) the training/test process is repeated many times to estimate a distribution of hold-out parameters. The central tendency of the hold-out parameter's distribution is taken for each of the subject factors and variable loadings. For example, a standard k-fold cross-validation would divide the subjects evenly into an 80–20 train/test split. The CCA would be fit, or trained, on 80% of the subjects. The trained parameters from 80% of the subjects would be applied to the 20% holdout. The canonical factors and variable loadings are compared between the train and test data to determine how well the model generalizes to the population.

To perform the repeated k-fold process, the dataset was first split into five equal groups of subjects. Then the CCA was fit five times using each combination of four of the groups to train the model, predicting the hold-out estimates for each of the splits. This creates a full set of hold-out estimates for every subject's factors ($F$) and loadings ($L$). This process is repeated 15,000 times, with each iteration creating a new random split of subjects into five groups before repeating the train/test

procedure. This creates a distribution of trained factors and loadings for every parameter in the model. The median value of these distributions indicates the most probable trained value for the parameter. In addition, the central tendency of these distributions is observed to quantify the standard error of the estimates. These median values of the permuted hold-out distributions are the factors and loadings used in evaluating the CCA (the factor values in Fig. 1, the loading values in Fig. 2). The repeated k-fold cross-validation approach allows the full dataset to be evaluated in the results, as each subject's factor scores and each variable's loadings are estimated from a model using unique observations to predict their estimate. This approach was necessary because the modeling of data with a CCA does not generalize to new observations well. The estimation of the CCA is based largely on deterministic matrix algebra, making the final results very sensitive to small perturbations in the input data. Selecting new training-test splits from the observations in a single k-fold cross-validation approach often resulted in different conclusions, while this repeated k-fold cross-validation consistently produced similar model estimates.

In addition, this cross-validation approach is robust to our data being evenly sampled across our outcome of interest, age. Any one random sample of subjects from this dataset would be likely to bias the training or prediction of a particular age. Further, it is not possible to use age as part of the identification of training subjects while still being able to use it as an unbiased outcome. By cross-validating across many random k-fold splits we are able to cross-validate our model without risking an unintended bias with age that may be present in any one split.

*Bootstrap test.* A bootstrap method was used to test the statistical significance of the correlations and loadings within the CCA models ($M_0$, $M_1$, and $M_2$). The bootstrap tests were implemented by assuming a null hypothesis of no correlation between the two variables (or parameters sets). For each of the various tests described in the "Results" section, a sampling distribution of the mean was generated by resampling the data (or model parameters) with replacement. For each bootstrap test, 10,000 samples of the data (or model parameters) were generated. The probability of the empirically observed correlation given the sampling distribution was reported ($p$). Gray shading in Supplementary Fig. 2i, j show examples of the resampling distributions generated by bootstrap under the null hypothesis of no correlation between variables.

*Building the average contribution of brain networks and behavior domains as plots and word clouds.* To summarize the brain networks and behavior domains that most strongly contribute to $M_1$, the variables within each dataset were averaged within predetermined modules and summarized with a scatter plot and word-cloud plot. The module for the brain network loadings was built by assigning the labels from the HCP-MMP (v1.0) atlas to a set of established functional network labels from ref. [93], referred to as $Y_{2011}$. Each of the 376 nodes in our networks was assigned to one of the seven functional networks in $Y_{2011}$; visual, somatomotor, limbic, ventral attention, dorsal attention, frontoparietal, and default-mode networks. In addition, the hippocampus and amygdala were kept separated and all the remaining subcortical structures were combined and reported as subcortical (i.e., pallidum, putamen, accumbens, thalamus, and caudate). This process generated a total of ten labels (words) representing the established functional brain networks of $Y_{2011}$, referred to as modules. The modules for the behavioral domain were determined from the CAN domains that the tasks were chosen to sample across (Memory, Language, Emotion, Attention, Motor, Clinical Scores, and Social). First, the individual variable loadings were taken from the cross-validated model. Next, the variables from each axis were combined within each module by averaging together all the positive values into a positive average and the negative values into a negative average. By creating separate averages for the positive and negative loadings, the prevalence of the modules (brain network or behavior) on each end of the axis was assessed. The error bars were computed by taking the average of the respective variables' standard errors from the cross-validated model. The left-hand size plots in Fig. 3c, d display the average variable loading within each brain network or behavioral module as a pair of points. The positive average (blue) and negative average (red) contribution of variables within the respective module are displayed as a pair of columns. Error bars show 2 units of the average standard error estimated by cross-validation. The right-hand side plots in Fig. 3c, d show word-cloud representations of the variable modules. These plots were created using the same average values (Fig. 3c, d left-hand side), scaling the font size by sum of the modules loadings. More specifically, the font size of each word was scaled by the average of the loadings within each module. The larger the font of the word, the larger the average for positive values (blue) or lower for negative values (red). For example, on the behavioral axis (Fig. 2a, *y*-axis), on average all the memory tasks contributed a lower score than the emotion tasks, hence the font size for memory is smaller than that for emotion in Fig. 3d.

*Computing the rich club of the network and its comparison to the CCA.* In order to compare the CCA model to the rich club (RC; ref. [103]), a global network was constructed by averaging individual networks across all subjects. A standard consensus threshold was applied after averaging by zeroing any connection not present in a simple majority of individuals (i.e., a connection was kept in the average matrix if the connection was available in more than 50% of the total number of subjects). Next, node degree was estimated on this averaged network

using the brain connectivity toolbox[91]. The node-degree estimates were normalized by subtracting the minimum and dividing by the maximum. A threshold was then applied to the node degree to identify the RC. We used results from previous work[103] and set the threshold of 14% to separate RC from periphery. This means that the top 14% of the nodes with the highest normalized node degree were assigned to RC. More specifically, 54 of 376 regions in the network were assigned to the rich club and the remaining 322 to the periphery—see color-coded ball-stick representations in Fig. 4. The size of the nodes in Fig. 4 was scaled by the magnitude of the CCA loading of each node.

After identifying the RC and periphery, the association between RC/Periphery and the CCA was investigated. The spearman rank correlation was estimated between the node degree in each node and CCA loadings. A bootstrap test was used to estimate the mean, standard error, and $p$-values reported in "Results". To visualize the correlations, a scatter plot comparing the CCA variable loadings and the participation coefficient of the RC is shown in Supplementary Fig. 4a. The points are color-coded for the rich club (blue) and periphery (gray). The ordinate of Supplementary Fig. 4a is the RC participation coefficient for each node. The participation coefficient was computed as the ratio of the number of connections from that node to other nodes in the same group (RC or periphery) to the number of connections from that node to the other groups[103]. The full list of CCA loadings in relation to the participation coefficient was displayed (Supplementary Fig. 4b) to show that the majority of RC connections also have relatively high loadings within the CCA as well.

**Representational similarity analysis (RSA).** We used representational similarity analysis (RSA; refs. [92,137]) to create a fingerprint of how the loadings of the variables in one data domain (brain) are associated with the loadings of the variables in the other data domain (behavior). We define factors as the estimated latent factors from the CCA analysis. The number of factors estimated is determined by the smallest number of tuned PCA parameters used during parameter tuning. Each variable has a loading for the factors estimated. We use a dissimilarity measure estimated between pairs of variable loadings to inspect how the variables interact with each other across all estimated factors. This allows us to observe how the individual observations within the domains of our CCA model, brain and behavior, interact with each other as they form the primary axes for all factors in the analysis. Typically, the extent of our inspection of the CCA model is using the cross-validated variable loadings ranked by their magnitude to indicate the strongest contributing variables from a particular domain (Fig. 2). However, there is currently no way to recover how the individual variables interact with each other. While the CCA determines an overall interaction between the separate datasets and the variables that most contribute to each factor, there was previously no description of how the individual elements interact with one another in this model. Access to this information would prove particularly useful to determine what relationships within the data may be capitalized on to estimate the canonical factors, as there are many established brain-behavior interactions within this dataset.

The dissimilarity matrix ($S$) is positive and block symmetric, with three distinct components representing the network-to-network, behavior-to-behavior, and brain-to-behavior dissimilarity. After building $S$, the elements of the matrix were sorted and organized into meaningful brain and behavioral domains. This organization allowed us to interpret values of $S$ in terms of functional domains for both brain and behavior instead of individual regions or tasks. Multiple individual brain regions in $\theta_B$ were sorted and uniquely assigned into the $Y_{2011}$ labels (ref. [93]; https://github.com/bcmcpher/cca_aging/blob/master/HCP_to_Yeo_assignment.tsv). The behavioral measurements in $\theta_b$ were mapped onto the seven major behavioral domains and averaged (Social, Emotional, Attention, Memory, Clinical, Language, and Motor). Results show that the CCA captures patterns of covariation between the two domains (brain and behavior). For example, the emotion domain is highly similar (low dissimilarity) both in the brain and behavior between the brain and behavioral contributions. Also, attention and memory are highly similar (low dissimilarity) as well. To contrast these results to the normalized data, see Supplementary Fig. 4i-k that shows the difference in range between the dissimilarity of the observed variables ($x$-axis) to the CCA estimates ($y$-axis).

To build the dissimilarity matrix all variables used in the analysis, we first use the variable loadings reconstructed from the cross-validated factors. To construct the RSA, we take the dissimilarity measure between every pair of variables using their recovered loadings for all factors. This is taken across both $L_1$ and $L_2$ (the loading matrices for brain and behavior, respectively). Each row represents the canonical loadings for every canonical factor for the variable and each column is the order of the estimated canonical factors. By taking the dissimilarity measure between each row, we get a measure of how similarly any two variables behave across all the estimated factors. The dissimilarity is able to be taken across domains, meaning how the brain and behavior variables interact within their respective factors can be inspected directly.

The first factor of the CCA represents the factor structure of each domain that creates the strongest correlation between the datasets, while each subsequent factor represents the next orthogonal configuration that has the next strongest correlation. By comparing the variable loadings of each factor between two variables we can represent how the variables interact with each other across every estimated canonical factor. We use this full vector of estimated factor scores to

determine the distance by calculating the dissimilarity between every unique pair of variables. A dissimilarity measure:

$$S_{\{M\}} = 1 - |\text{corr}_{\{x,y\}}| \tag{4}$$

where $x$ and $y$ are any pair of variable loadings in an $[n \times 1]$ vector. Each vector consists of the $n$ loadings for each canonical factor estimated in the model for each variable. The correlation between these vectors would represent how similarly the variables contribute to each of the estimated canonical factors. Every unique combination of $x$ and $y$ created a symmetric matrix of size $[710 \times 710]$ with the respective dissimilarity value between every variable estimated. The values represent how dissimilar any pair of variables behave across every estimated factor. The higher the score between the variables, the more independently they interact across all the estimated canonical factors. Likewise, the lower the score between the variables, the more similarly they behave across factors. To summarize this information, we sorted these measures into established behavior domains and brain networks. This better reveals how variables within a specific domain behave with variables from another domain. From this sorted matrix, we estimate the modules by averaging each value within a module into a single value. This creates a matrix of size [modules × modules] with the mean and standard deviation of dissimilarity values for every variable within that group. These values represent on average how the behavior domains and brain networks interact with each other across all the factors. However, the most important interaction for our purpose is the interaction between the brain-behavior averages, or how well the behavior domains correspond to the structural connectivity within established functional networks (Fig. 4e).

To do so, we computed the loadings by correlating the original normalized data and the CCA factors for each canonical correlate estimated (columns of $D_{1,2}$ with columns of $F_{1,2}$, respectively). Two matrices were generated with this approach, one for the behavioral and one for the brain network loadings ($L_1$ and $L_2$). These loadings were then stacked into a matrix where each row is a variable and each column is the corresponding canonical axis (Supplementary Fig. 4a) and used to compute the dissimilarity matrix between all brain regions and behavioral tasks. This creates a symmetric matrix where the pairwise dissimilarity distance between all rows of the stacked matrices was computed using Eq. (4) (Fig. 3a; ref. [92]). The axes are the size of the number of variables included in the model and a value for each pair of variables indicating how similarly they behave across all canonical factors.

*Chord plot.* To better visualize the association between brain and behavior we transformed the brain-behavior dissimilarity value back to correlations: $r = 1 - S_M$. These values were then used for a modified chord plot visualization. For this plot, the $r$ values were normalized by rescaling them to better show the brain-behavior interactions. First, $r$ was normalized between 0 and 1. Second, $r$ values below the top 75th percentile were eliminated (Supplementary Fig. 4h-j). Third, $r$ was multiplied by 1000 and squared to visually emphasize any difference in values. The chord plots represent the strength of the association between the CCA loadings of the brain networks and behavioral domains. Two aspects of this plot inform our analysis. (A) Each wedge of the chord shows the total association in a domain; the larger the wedge for a particular domain the larger the contribution to the CCA loadings. (B) The thicker the bands connecting a behavioral domain and a brain network, the stronger the association between the loadings in those variables.

*Evidencing the effect of age to the CCA by computing the difference in RSA between $M_1$ and $M_2$.* To determine the impact of age on the brain-behavior relationship, we repeated the whole cross-validated CCA procedure, but first age was regressed along with the other covariates before the CCA was performed, referred to as $M_2$ in the text. This effectively removes the association with age from the subsequent analyses as described in "Results". When the RSA analysis is repeated with $M_2$, the resulting dissimilarity modules reflect the brain-behavior interaction with the variability of age removed. What is left represents the difference between models with and without age. By taking the difference between the $S_1$ and $S_2$ we are able to show the impact of age on the brain-behavior relationship ($S_d$). The difference between these modules is the impact of age on the connection between these scores.

### Statistics and reproducibility

*Model selection.* Cross-validation was used to select the best CCA model. The mean model parameters across 10,000 cross-validation thought was selected as the dominant model of CCA and used for primary analysis.

*Statistical significance.* Bootstrap methods were used to test the statistical significance of the correlations and loadings within the CCA models ($M_0$, $M_1$, and $M_2$). The bootstrap tests were implemented by assuming a null hypothesis of no correlation between the two variables (or parameters sets). For each of the various tests described in the "Results" section, a sampling distribution of the mean was generated by resampling the data (or model parameters) with replacement. For each bootstrap test, 10,000 samples of the data (or model parameters) were generated. The probability of the empirically observed correlation given the sampling distribution was reported ($p$).

*Reproducible network neuroscience data preprocessing.* As part of this work, a fully automated and reproducible processing pipeline was created. The pipeline is composed of seven composable web services, called Apps (Table 1). The data processing pipeline removed artifacts, performed fiber model fitting using a recent robust method (see ref. [96] and bl.app.68) and generates whole-brain tractograms using a novel pipeline called Reproducible Anatomically Constrained Ensemble Tracking (RACE-Track; bl.app.101). The network generation process combines regions of interest from an atlas (bl.app.23) to build connectivity matrices. The data preprocessing Apps can be accessed for reuse at brainlife.io/apps and by accessing the data and processing record: brainlife.io/pubs: https://doi.org/10.25663/brainlife.pub.21.

## Data availability

The source data are provided by the Cambridge Aging Neuroscience Project https://camcan-archive.mrc-cbu.cam.ac.uk/. Brain data derived as part of this project and used as features for all the analyses are available on brainlife.io/pubs:https://doi.org/10.25663/brainlife.pub.21. Source data from the plots is distributed in Supplementary Data 1. All other data are available from the corresponding author upon reasonable request.

## Code availability

Code is available on github at https://github.com/bcmcpher/cca_aging.

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

## Acknowledgements

This research was supported by NSF IIS-1912270, NSF IIS-1636893, NSF BCS-1734853, Microsoft Faculty Fellowship to F.P., NIH 5T32MH103213-05 to William Hetrick. We thank Soichi Hayashi, Brad Caron, and Josh Faskowitz for contributing to the development of brainlife.io, Dan Stanzione, William. J. Allen, and James Carlson for support at TACC supercomputing center, and Craig Stewart, Robert Henschel, David Hancock, and Jeremy Fischer for support with jetstream-cloud.org (NSF ACI-1445604).

## Author contributions

B.M. processed and analyzed data; implemented the data-analysis software; conceptualized the analyses; interpreted results; wrote the manuscript. F.P. conceptualized and developed the study, figures, analyses, and interpreted results; supervised all aspects of the research; wrote the manuscript.

## Competing interests

The authors declare no competing interests.
