## [Peer Review File · Communications Biology]

Reviewers' comments:

Reviewer #2 (Remarks to the Author):

This work by McPherson & Pestilli examines the multivariate relationships between structural brain networks, behavior, and healthy aging in a large dataset spanning the lifespan. First, the authors show that both graph theoretic measures of structural brain networks and behavioral measures vary by age as hypothesized by previous works. Then, they use canonical correlation analysis to describe the multivariate mappings between brain and behavioral measures in this dataset. They find that much of variance observed along the first canonical axis is correlated with age (M1) and that controlling for age (M2) reduces the amount of variance to be explained. Finally, they show how the variable loadings from these models relate to one another using representational similarity analysis. While this work is important and well motivated, there are several questions and concerns that I have with the current manuscript.

1) Modeling age-related differences. While I do find it compelling that M2 (the model that controls for age) explains less of the variance in the data than M1, it's not clear to me that this necessarily means the loadings/multivariate relationship utilized in M1 are purely age-related. Particularly since these are not longitudinal data it seems challenging to claim that individual differences are not contributing to this model. I am also curious as to why the univariate relationships between brain/behavior and age were quadratic (e.g., highest node degree), but the multivariate relationships along CA1 were linearly related to age?

2) Reliability of the loadings/weights. Since several figures depict the features/loadings, it would be useful to know how variable these weights are across folds/permutations. If these weights are highly variable it may not be warranted to look into the specific variable loadings. In addition, even though the feature loadings in M2 were not significant, it would also be useful to compare the variable loadings between M1 and M2. This might provide more evidence that age-related differences and individual differences are separate (or not).

3) Chord plots and representational similarity analysis. It is not exactly clear what question representational similarity analysis is meant to address here. It seems like a consideration of the magnitude of the loadings would be needed--variables with high similarity, but non-significant loadings may not be of interest? It is not exactly clear what the chord plots are depicting. Given the dissimilarity plot in Figure 4b, I would have expected the opposite in 4c (e.g., Social is the least dissimilar across most brain networks, so should have the strongest associations?) It would also be helpful if the authors would comment on the differences between M1 and M2. Why do the variable loadings in M2 appear to have more structure (especially the behavioral measures)? Lastly, it is difficult to see the differences in size between each "chord" in the chord plot.

4) Discussion. While the authors successfully argue the importance of multivariate approaches, population neuroscience, and reproducible analyses, there should be more discussion of what these results reveal about healthy aging. It is not clear what has been added to our understanding of the mechanisms by which brain networks change across the lifespan.

5) Word clouds. I found the word clouds in Figure 3c and 3d to be difficult to understand. For example, in 3c why are Limbic and DMN in red larger than Hippocampus in blue? This does not seem to match Figure 3a. It might be simpler to group the behavioral/brain loadings in Figure 3a/b by these domains/networks.

Reviewer #3 (Remarks to the Author):

Thank you for the opportunity to review "A single mode associates global patterns of brain network structure and behavior across the human lifespan" by McPherson and Pestilli. The manuscript describes a canonical correlation analysis of diffusion-estimated structural connectivity and behavioral measures in the Cam-CAN sample. The authors find a single dominant canonical variate/mode that is reproducible and is correlated with age.

The manuscript is clear and addresses an interesting question. The methods are rigorous and executed well. My comments are mostly minor:

(1) The brain side of Figure 3 could include additional maps, such as the information in panel a plotted on a brain surface.

(2) On a related note, it would be interesting to know whether the mean loadings for each brain region correlate with their degree, participation, etc. Do members of the putative "rich club" have greater loadings than the periphery?

(3) The authors cross-validate the correlation between CCA-estimated subject scores, but do the weights resemble each other across resamplings?

(4) The authors could assess how specific network affiliation is in panel 3c by computing mean loadings in each RSN and simply showing this information as a bar. Additional permutations or spin tests could be used to confirm specificity.

(5) It was not clear how the loading correlation matrix was constructed for RSA. What are the actual observations that the correlation is computed over?

(6) "For each of the various tests described in the results section, a null distribution was generated by resampling the data (or model parameters) with replacement" (pg. 14). I would argue this is the sampling distribution, not the null distribution.

Reviewer #2 (Remarks to the Author):

This work by McPherson & Pestilli examines the multivariate relationships between structural brain networks, behavior, and healthy aging in a large dataset spanning the lifespan. First, the authors show that both graph theoretic measures of structural brain networks and behavioral measures vary by age as hypothesized by previous works. Then, they use canonical correlation analysis to describe the multivariate mappings between brain and behavioral measures in this dataset. They find that much of variance observed along the first canonical axis is correlated with age (M1) and that controlling for age (M2) reduces the amount of variance to be explained. Finally, they show how the variable loadings from these models relate to one another using representational similarity analysis. While this work is important and well motivated, there are several questions and concerns that I have with the current manuscript.

Thank you for the nice summary and for agreeing on the importance of the work.

1) Modeling age-related differences. While I do find it compelling that M2 (the model that controls for age) explains less of the variance in the data than M1, it's not clear to me that this necessarily means the loadings/multivariate relationship utilized in M1 are purely age-related. Particularly since these are not longitudinal data it seems challenging to claim that individual differences are not contributing to this model.

Thanks for the comment. Indeed, individual differences across age is what is best predicted by the model. Indeed, we believe that the contributions of all the variables we used across-the-lifespan is what makes M_1 work so well.

In response to the reviewer, we changed the title of the section on Page 4. The new title is "A single-mode of covariation relates individual differences in structural networks and behavior with subjects' age," and it is meant to highlight the fact the results capture individual differences.

Furthermore, we edited the results section on Page 6 to clarify this issue. We have added the following sections at the end of Page 6 in Results:

These results show that the quadratic trends in the two data domains are well captured by the linear trend of the CCA. More explicitly, the quadratic trends for individual variables across the lifespan (as shown in **Figure 1e, f**) in the two data domains (behavior and brain networks) are well coupled by CCA into a linear trend. As a result, the CCA result is strongly associated with the participants' age.

To summarize this main result, we implemented a hypothesis-driven approach to CCA using cross-validation and tested our hypothesis of an association between age and the CCA model. More specifically, we tested the degree to which a single variable, participants' age, predicted (in cross-validation terms) the variability in the linear combination between hundreds of variables from brain and behavioral measures. The results show that a major portion of the variability in the CCA model is effectively associated with age. This result does not mean that other variables were not associated with the model, indeed they were as the CCA model is significant and correlation in CA_1 is strong.

We judge this to be the main result of our work.

I am also curious as to why the univariate relationships between brain/behavior and age were quadratic (e.g., highest node degree), but the multivariate relationships along CA1 were linearly related to age?

Thanks this is an important point. The new section on Page 6 and reported above addresses this comment. In a nutshell, the CCA is a linear combination of variables with non-linear (quadratic) trends in the two data domains, so that the resulting trend of the CCA model is linear. The CCA shows optimal combinations of the variables to maximize the relationship between the domains - it does not reflect the trends observed with each individual variable.

2) Reliability of the loadings/weights. Since several figures depict the features/loadings, it would be useful to know how variable these weights are across folds/permutations. If these weights are highly variable it may not be warranted to look into the specific variable loadings.

We appreciate the reviewer making this suggestion. Indeed, a great deal of our work went into establishing an approach to cross-validate the model testing so to estimate for example the variability in the final model parameters.

In response to the reviewer's comment, we have changed Figure 3a and b and Supplemental Figure 3b and c. The new plots report the error bars (± 2 s.e.m.) to provide a visualization of the variability of the weights. We are pretty happy with the result as the variability in the weights is as small as one could predict for this type of analysis.

In addition, even though the feature loadings in M2 were not significant, it would also be useful to compare the variable loadings between M1 and M2. This might provide more evidence that age-related differences and individual differences are separate (or not).

Thanks for asking for this additional report. Two responses to this comment:

First, we have now changed the report of M₂ in Supplemental Figure 3 to match panel-for-panel the report of M₁ from Figure 3. Similarly to the revised Figure 3a and b, we now also report the magnitude of weights (mean) and ± 2 s.e.m of the weights for CA₁ of M₂ in Supplementary Figure 3. The following section and table was added the the supplementary document:

The reader can compare the loadings for M₁ and M₂ by comparing **Figure 3a and b** and **Supplemental Figure 3b and c**. **Figure 3a and b** reports the loadings for M₁ CA₁. **Supplementary Figure 3b** reports the loadings for M₁ CA₂ and **Supplementary Figure 3c** reports the loadings for M₂ CA₁. See the table below for additional clarification.

	CA ₁	CA ₂
M ₁	Figure 3a and b	Supp Figure 3b
M ₂	Supp Figure 3c	not reported (extremely small loadings)

Supplementary Table 1. References to Figures containing Models and CCA axes.

The results show that whereas the weights for M₁ CA₁ are large and reliable, the weights for both M₁ CA₂ and M₂ CA₁ are much smaller and more variable. This comparison supports the hypothesis that a single axis predicting the quadratic trends across the lifespan (i.e., M₁ CA₁); hence when the CCA model is built without removing the

participants' age. Opposite to that, if the participants' age is removed as in the case of M2 the CCA model fails in predicting a substantial portion of the variance in the data from the two domains.

Second, we would like to point out that the suggestion of the reviewer to compare M_1 and M_2 directly is precisely what we do in the last analysis with the RSA. We believe this point was not made clear in the first submission, we have now revised text and figures to make sure this critical point is evidenced. As the reviewer suggests by comparing M_1 and M_2 we really can get at the effect of aging. Thank you for clarifying this, the next section here will respond to this issue further.

3) Chord plots and representational similarity analysis. It is not exactly clear what question representational similarity analysis is meant to address here.

This section was rewritten to help clarify our goals. Our initial draft had an oversight in how the figures were labeled, which led to some confusion. Figure 5 does not just show an RSA analysis of M_1 , but instead a comparison (a subtraction) of M_1 and M_2 . For this reason the analysis and Figure 5 shows the difference between the RSA built using M_1 and the RSA built using M_2 .

In response, we have now updated Figure 5, the title of the relevant section so to explicitly mention the "Difference in Dissimilarity" and the text describing the figure.

As the reviewer noted in the previous comment, a comparison of M_1 and M_2 , help the analysis to highlight associations specific to age and not just the individual variables.

It seems like a consideration of the magnitude of the loadings would be needed--variables with high similarity, but non-significant loadings may not be of interest?

We appreciate the point raised by the reviewer. Indeed, in the sections preceding the RSA analysis, we have clarified that only CA_1 is statistically significant for M_1 . All the remaining axes CA_{2-38} are non significant and most of their some variables have non-significant loadings. So this is a fair point.

Because significance is an arbitrary threshold, in the RSA analysis we do not focus simply on the significance of individual axes. Instead, we focus on the total variance explained by the CCA model. For that we use all axes of CA_1 and subsequent and use the RSA method to highlight global patterns of association between variables domains. The approach of applying a RSA to the CCA loadings is a unique proposal that appears to provide new insights to a popular model. While it may not be adopted as is, we believe this approach is a useful starting place for further innovation by the field.

In response, we have heavily revised the sections of the text regarding the RSA to clarify how the model was built.

It is not exactly clear what the chord plots are depicting. Given the dissimilarity plot in Figure 4b, I would have expected the opposite in 4c (e.g., Social is the least dissimilar across most brain networks, so should have the strongest associations?)

In response to the comment, we have revised this section to better clarify the goal of the plots and how the data was used to construct them. Below an extract from the text on Page 9:

The results were visualized using a modified chord plot that allowed us to show how multiple associations between brain and behavior load onto M_1 simultaneously (**Supplementary Figure 5e**; see **Methods**). The chord plot was generated by

thresholding the RSA values in *btn* the top quartile. Two aspects of the chord plots shown in **Supplemental Figure 5e** are of interest. First, the number of domains (or networks) that each network (or domain) contributed to is described by the size of the peripheral segments for each network and domain. The larger the size of the segments, the more contributions of a network (domain) to the various other domains (networks). Second, the associations between each functional network and behavioral domains are described by individual chords (if a chord exists two domains or networks interacted in the CCA).

Lastly, it is difficult to see the differences in size between each “chord” in the chord plot.

Thanks, indeed looking at the width of the chord would be of interest. As described above we focussed on the more dominant patterns by looking at which variables are associated (share a chord) with one another. The two primary aspects of interest of the chord plot are the size of the wedges in the periphery and the size of presence or absence of a chord between two variables.

We have changed the text to better emphasize what the plot means and how it should be read.

It would also be helpful if the authors would comment on the differences between M1 and M2. Why do the variable loadings in M2 appear to have more structure (especially the behavioral measures)?

That is a great point. M_2 has the role of age in the association removed. Because of this is it possible to infer that the associations between the variables stand out more clearly in M_2 . Yet, we note the model explains very little of the variance in the data - neither the canonical axis' correlation with age nor the variable loadings not statistically significant.

4) Discussion. While the authors successfully argue the importance of multivariate approaches, population neuroscience, and reproducible analyses, there should be more discussion of what these results reveal about healthy aging. It is not clear what has been added to our understanding of the mechanisms by which brain networks change across the lifespan.

We have added a paragraph in the discussion (Page 11) to address this concern. We argue that even though the patterns of association we report are global and cannot be specifically directed to one variable or another, it is an important contribution to demonstrate the level of co-variability in the changes in networks and behaviors across age.

5) Word clouds. I found the word clouds in Figure 3c and 3d to be difficult to understand. For example, in 3c why are Limbic and DMN in red larger than Hippocampus in blue? This does not seem to match Figure 3a.

We thank the reviewer. Indeed, we had originally normalized the word clouds for visualization purposes by separating the positive and negative contributions. In this way, the scaling of the words was easier to read (i.e., the positive and negative values were normalized independently). However, we recognize this caused confusion for the reviewer as it removed the relative scaling between the positive and negative contributions.

In response to the reviewer's comment we have now changed the scaling of the text in the word clouds. Both the positive and negative contributions are scaled by the same factor: the average loading within the positive and negative contributions. To normalize across the negative and positive contributions, we computed the absolute value of the average loadings and used that to normalize the font size.

It might be simpler to group the behavioral/brain loadings in Figure 3a/b by these domains/networks.

Thank you for suggesting this. We tried to create the visualization as suggested but we found that it was not as clear as we hoped for. Figure 4b is a similar figure to what is requested. The lack of readability in the individual variables made us reconsider its broader utility. While less explicit, the average contributions within each domain are far easier to interpret and explain much of the same information.

We responded to this comment by changing the scaling of the words in the cloud as described above. To respect the new changes, we also updated the methods section, specifically the section pertinent to the word cloud generation (Page 17).

Reviewer #3 (Remarks to the Author):

Thank you for the opportunity to review “A single mode associates global patterns of brain network structure and behavior across the human lifespan” by McPherson and Pestilli. The manuscript describes a canonical correlation analysis of diffusion-estimated structural connectivity and behavioral measures in the Cam-CAN sample. The authors find a single dominant canonical variate/mode that is reproducible and is correlated with age.

The manuscript is clear and addresses an interesting question. The methods are rigorous and executed well.

We thank the reviewer for appreciating the clarity, rigor in methods and execution.

We believe this is indeed a key aspect of our contribution as the way to approach CCA for understanding brain and behavior has been the matter of debate in scientific papers as well as outside (<https://www.discovermagazine.com/mind/scarred-brains-or-shiny-statistics-the-perils-of-cca>).

We believe our approach that uses a key variable of interest (age in our case) to test specific hypotheses on its role in building a full CCA model across multiple variables and data domains can be helpful to other researchers. Indeed we have gone a long way to provide access to the new approach we introduce here. We have now published the open services used to process the data as reported in our work on brainlife.io, the services can take raw data and generate structural connectivity matrices as used in our work. Furthermore, we have published a full Matlab tool box compatible with the Brain Connectivity Toolbox that allows taking the brain networks generated on brainlife.io and behavioral data associated with the same subjects and build CCA models, cross validate them and test hypothesis related to individual variables, just like we did we age: https://github.com/bcmppher/cca_aging

We hope the contribution to the methods for CCA will increase the impact of the present work.

My comments are mostly minor:

(1) The brain side of Figure 3 could include additional maps, such as the information in panel a plotted on a brain surface.

We appreciate this suggestion. We agree that a representation of the variable loadings from the CCA axis on the brain may better communicate the general extent of the findings to the reader.

In response, we have now added an inset in Figure 3a. The new inset displays the CA₁ loadings mapped to the cortical surface. We have also edited the text on Page 7 to describe the global patterns shown by the brain maps in relation to the various brain areas. The results are consistent with the findings reported in a later paragraph in the same section using the Yeo et al. 2011 Atlas. We remind the reader of those sections.

(2) On a related note, it would be interesting to know whether the mean loadings for each brain region correlate with their degree, participation, etc. Do members of the putative “rich club” have greater loadings than the periphery?

We thank the reviewer for this suggestion. We put a lot of efforts into generating new insights from this network neuroscience approach. We find the results meaningful and interesting and are grateful for the help.

Our response involved adding a new:

- Figure. See new Figure 4 Page 7.
- Section in the Methods See Page 17.
- Section in the Results. See Page 7.

In a nutshell, we performed a new analysis exploring the relationship between the average network’s rich club to the CA₁ brain loadings. To do so, we:

1. Inspected and reported the correlation between the average (mean across subjects) network’s node degree and the CA₁ variable loadings, independently for rich-club core and periphery.
2. Inspected the average variable loadings across network nodes within the rich-club core or periphery.
3. (Supplemental) Estimated the participation coefficient for each node and compared to the loadings in a scatter plot.
4. (Supplemental) Reported the list of rich club members nodes and compared the ranked membership of the network nodes to the rich club to the CCA loadings.

We would again like to thank the reviewer for this suggestion. While we did not find a statistically significant difference distinguishing the average loadings for the rich-club core and periphery, we do find a significant correlation between the CA₁ loadings and node degree (the measure that determined the rich club core or periphery membership). So, we believe this additional analysis inspired by the reviewer is a valuable addition to our overall findings.

(3) The authors cross-validate the correlation between CCA-estimated subject scores, but do the weights resemble each other across resamplings?

Thanks for the suggestion. We note this is a comment similar to one made by Reviewer #2.

A large goal of our work was precisely to develop a framework based on cross-validation to produce a reliable and reproducible analysis using CCA. So we regret not having reported the error bars in the first submission.

In response to the reviewer’s comment, we have changed Figure 3a and b and Supplemental Figure 3b and c. The new plots report the error bars (± 2 s.e.m.) to provide a visualization of the variability of the

weights. We are pretty happy with the result as the variability in the weights is as small as one could predict for this type of analysis.

(4) The authors could assess how specific network affiliation is in panel 3c by computing mean loadings in each RSN and simply showing this information as a bar. Additional permutations or spin tests could be used to confirm specificity.

We appreciate the reviewers' feedback on this figure.

In response to the reviewer, we have added new plots to **Figure 3c** and **d**. The plots (left-hand side of the word clouds) show the variability of the estimates requested by the reviewer with symbols and error bars (± 2 s.e.m.).

(5) It was not clear how the loading correlation matrix was constructed for RSA. What are the actual observations that the correlation is computed over?

This point was also raised by Reviewer 2. We have revised the text describing the RSA substantially. This involved changes to the Results, Methods, Discussion as well as supplementary material. All new changes in the relevant section are marked in blue. We hope the new text will clarify the goals and approach for the RSA.

(6) "For each of the various tests described in the results section, a null distribution was generated by resampling the data (or model parameters) with replacement" (pg. 16-17). I would argue this is the sampling distribution, not the null distribution.

We thank the reviewer. We have revised the text to better clarify the resampling approach. Page 16 of the revised manuscript. We avoid calling it a null distribution and instead refer to it as the sampling distribution.

REVIEWERS' COMMENTS:

Reviewer #2 (Remarks to the Author):

I thank the authors for responding to my comments. Their clarifications were of much help. I believe the manuscript is acceptable in its current form.

Reviewer #3 (Remarks to the Author):

The authors have comprehensively addressed my concerns and I recommend publication.